

# Dynamic assessment of the effectiveness of digital game-based literacy training in beginning readers: a cluster randomised controlled trial

Toivo Glatz[1,2,3], Wim Tops[4], Elisabeth Borleffs[1,5], Ulla Richardson[6], Natasha Maurits[3,7], Annemie Desoete[8,9] and Ben Maassen[1,3]

[1] Center for Language and Cognition (CLCG), Faculty of Arts, University of Groningen, Groningen, Netherlands
[2] Institute of Public Health, Charité–Universitätsmedizin Berlin, Corporate Member of Freie Universität Berlin and Humboldt-Universität zu Berlin, Berlin, Germany
[3] Behaviour and Cognitive Neuroscience (BCN), University Medical Center Groningen (UMCG), University of Groningen, Groningen, Netherlands
[4] School of Educational Studies, Universiteit Hasselt, Hasselt, Belgium
[5] Department of Child and Adolescent Psychiatry and Psychotherapy, University Hospital of Psychiatry, University of Zurich, Zurich, Switzerland
[6] Centre for Applied Language Studies, University of Jyväskylä, Jyväskylä, Finland
[7] Department of Neurology, University Medical Center Groningen, University of Groningen, Groningen, Netherlands
[8] Department of Experimental Clinical and Health Psychology Ghent University, Gent, Belgium
[9] Artevelde University College of Applied Sciences, Gent, Belgium

Corresponding author
Toivo Glatz, toivo.glatz@charite.de

## ABSTRACT

In this article, we report on a study evaluating the effectiveness of a digital game-based learning (DGBL) tool for beginning readers of Dutch, employing active (math game) and passive (no game) control conditions. This classroom-level randomized controlled trial included 247 first graders from 16 classrooms in the Netherlands and the Dutch-speaking part of Belgium. The intervention consisted of 10 to 15 min of daily playing during school time for a period of up to 7 weeks. Our outcome measures included reading fluency, phonological skills, as well as purpose built in-game proficiency levels to measure written lexical decision and letter speech sound association. After an average of 28 playing sessions, the literacy game improved letter knowledge at a scale generalizable for all children in the classroom compared to the two control conditions. In addition to a small classroom wide benefit in terms of reading fluency, we furthermore discovered that children who scored high on phonological awareness prior to training were more fluent readers after extensive exposure to the reading game. This study is among the first to exploit game generated data for the evaluation of DGBL for literacy interventions.

## INTRODUCTION

Adequate early literacy instruction and well-developed literacy skills are indispensable for a child's academic success and future career. It is therefore important to know how we can

improve teaching methods and accurately monitor reading progress. Digital game-based learning shows potential in that it has a range of benefits over traditional (offline) teaching methods as it offers a multimodal learning environment to improve the engagement and learning of students (*Potocki, Ecalle & Magnan, 2013*). Such games also provide immediate feedback for improved learning, can adapt to individual learners depending on their responses, and are highly motivating for the players (*e.g.*, *Desoete et al., 2016*). In addition, they allow researchers to monitor the players' individual development of accuracy and response times over time in task-relevant contexts and to acquire valuable longitudinal game data of large groups of participants (*e.g.*, *Praet & Desoete, 2014*; *Puolakanaho & Latvala, 2017*). All this makes digital game-based learning a natural choice when investigating potential early recognition and remediation of reading difficulties in children.

## Reading impairment

Developmental dyslexia or specific learning disorder in reading, henceforth dyslexia, is a developmental disorder characterised by persistent difficulties in word recognition (reading) and/or spelling (DSM-5, *American Psychiatric Association, 2013*). These difficulties are not caused by a general cognitive delay or by a hearing or vision impairment. Depending on a narrow or wider definition of poor reading proficiency, dyslexia affects around 4 to 12% of children across languages (*e.g.*, *Schulte-Körne et al., 1998*; *Schumacher et al., 2007*). Language and orthography both play an important role in reading (*Borleffs et al., 2017*), with the prevalence of dyslexia differing across languages depending on their characteristics (*Bergmann & Wimmer, 2008*; *Ziegler & Goswami, 2005*). Because of differences in the mapping of grapheme-phoneme correspondences, the developmental trajectory and nature of the reading problems may also differ between languages with regular and less regular orthographies (*Seymour, Aro & Erskine, 2003*; *Bergmann & Wimmer, 2008*; *Ziegler & Goswami, 2005*; *Vaessen et al., 2010*).

Dyslexia has a multifactorial aetiology in that it is associated with a range of genetic, environmental, and cognitive risk factors rather than with a single cause (*Pennington, 2006*). If one parent or sibling has dyslexia, the incidence rate rises to around 45%, indicating a familial risk (for a review, see *Snowling & Melby-Lervåg, 2016*). However, genetic risks do not operate in isolation. Reading is also influenced by environmental factors such as those related to parental socioeconomic status and their interaction with genetic factors (*Mascheretti et al., 2013*). Moreover, reading problems often seem associated with below average performance on specific cognitive and behavioural factors. The most prominent factors are letter knowledge, phonological awareness, and rapid automatised naming (of letters, digits, and familiar objects). Early performance on these skills predicts both reading accuracy and fluency (*van der Leij et al., 2013*; *Lyon, Shaywitz & Shaywitz, 2003*; *Lyytinen et al., 2004*; *Lyytinen et al., 2009*; *Moll et al., 2014*). However, certain cross-linguistic variability exists with respect to the relative weight of each of the cognitive and behavioural predictors of reading acquisition (*Landerl et al., 2013*; *Ziegler et al., 2010*). Letter knowledge is most predictive in Finnish with its extreme letter-sound consistency (*Lyytinen et al., 2009*); rapid automatised naming is the best long-term

predictor in German (*Brem et al., 2013*); and letter knowledge, rapid automatised naming, and phonological awareness are important predictors in Dutch (*van Bergen et al., 2012*).

For letter knowledge, we can further distinguish between letter-name knowledge and letter-sound knowledge, which rarely coincide in English orthography (*e.g.*, 'a' in 'bark', 'w' in 'wrong'). The assessment of letter-sound knowledge is the more suitable predictor of word reading in young children at the (very) beginning of reading instruction. However, letter-sound knowledge quickly reaches ceiling and therefore has limited use for long-term prediction. By adding time pressure to the letter-sound knowledge task, a more challenging task yielding a more sensitive measure can be created. Instead of only measuring the availability of letter-sound associations, timed letter-sound knowledge measures the fluency by which letter-sound associations can be retrieved from long term memory which is a proxy for the more general reading-related multimodal audio-visual information processing skill (*Blomert, 2011*; *Hahn, Foxe & Molholm, 2014*).

Given the multifactorial aetiology of dyslexia and early predictors such as poor letter-sound knowledge, phonological awareness, and rapid automatised naming, the question arises whether timely training of these skills might help remediate or even prevent reading difficulties. While benefits of training letter-sound knowledge and phonological awareness have been shown, rapid automatised naming seems more an individual characteristic than a trainable skill (*Brem et al., 2013*; *Landerl et al., 2013*; *Landerl & Wimmer, 2008*; *de Jong & Vrielink, 2004*; *Wolff, 2014*). In a position article, *van der Leij (2013)* summarised that an early intervention targeting reading precursors gives a head start but that training effects do not transfer to reading beyond the end of first grade. Therefore, effective interventions should not only start early but also be adapted to long-lasting educational needs.

## The GraphoGame framework

One promising way to deliver such an extended training is by computerised gaming (*Chambers et al., 2008*; *Richardson & Lyytinen, 2014*). GraphoGame is just such an adaptive computerised game targeting the training of reading-related skills, originally designed for Finnish with its transparent orthography (*Lyytinen et al., 2009*).

In transparent languages, letter-sound correspondences, and the speed by which these correspondences can be processed, were found to be the most consistent predictors for proficient reading (*Lyytinen et al., 2009*). Therefore, the first version of the game aimed to boost grapheme-phoneme correspondence knowledge in beginning readers by establishing solid and reliable connections between graphemes and phonemes (for a review, see *Richardson & Lyytinen, 2014*). Considering the empirical evidence for long-term development of word reading fluency, GraphoGame proved to work well for Finnish (*Saine et al., 2010*, *2011*). To make GraphoGame usable and effective in other, less transparent languages, more recent versions of the game do train grapheme-phoneme correspondences and phonological awareness, in addition to syllable and word reading fluency and spelling (*Richardson & Lyytinen, 2014*). So far, experimental studies using GraphoGame have been conducted in over 20 different languages, using different methodology and yielding mixed results.
A recent meta-analysis revealed that although the average gain in word reading fluency across 19 GraphoGame studies was close to zero, a few of the larger studies did show positive effects on reading fluency especially for at-risk readers, and many studies showed benefits for reading-related skills (*McTigue et al., 2020*). Interpretation remains complicated due to large differences between studies in various methodological aspects, including selection criteria for participants (poor performers below a certain cut-off score, teacher recommendation, genetic risk for reading impairment, entire classrooms), age of participants (5 to 10 years), the number and types of control groups (none, active and/or passive controls), sample sizes per group ($N = 10$ to 185), task time (1 to 8 h), training implementation (during school or at home, with or without adult engagement), training period (1 to 28 weeks), and type of language (ranging from transparent orthographies like Finnish to opaque orthographies like English).

Regarding reading fluency, for example, *Saine et al. (2010)* found that playing GraphoGame in Finnish improved reading fluency in first graders at-risk of reading impairment to the level of typically-developing peers by second grade. For reading-related skills, increased performance was observed in the domains of letter-sound knowledge in Dutch, Finnish, second language learners of English and French (*Blomert & Willems, 2010*; *Lovio et al., 2012*; *Patel et al., 2018*; *Ruiz et al., 2017*), phonological processing in Finnish (*Lovio et al., 2012*) and first and second language learners of English (*Patel et al., 2018*; *Kyle et al., 2013*), and sublexical skills in the form of syllable reading in versions made for German in Austria (*Huemer et al., 2008*) and Finnish (*Heikkilä et al., 2013*). This confirms the promising traits of the GraphoGame framework, but the presence of large methodological variations makes it challenging to grasp the characteristics that make up a successful GraphoGame intervention.

## The current study

The aim of the current study was to evaluate the effectiveness of a newly created version of GraphoGame for the semi-transparent Dutch orthography as compared to the earlier results described for a highly transparent language like Finnish (*e.g.*, *Saine et al., 2010*, *2011*) and more opaque languages like French (*Ruiz et al., 2017*) and English (*Kyle et al., 2013*; *Worth et al., 2018*). Our secondary aims were to investigate whether characteristics of participants and the intervention itself modulate the response to GraphoGame-NL intervention and what impact different forms of assessment have in the evaluation of intervention effects.

**Question 1:** Do children playing GraphoGame-NL for up to seven weeks at the beginning of first grade show a larger response to intervention in word reading fluency, phonological awareness, and/or letter-sound knowledge compared to children playing a control game and children not playing at all, while all groups follow the conventional classroom curriculum? *Hypothesis 1*: In beginning, GraphoGame-NL improves reading-related skills like letter-sound knowledge and phonological awareness to a bigger extent than word reading accuracy and speed.

**Question 2:** Do certain subgroups of children benefit more from GraphoGame-NL exposure than others? *Hypothesis 2*: Children with a particular risk factor, such as familial

risk for dyslexia, younger participants, speaking a foreign language at home, or children who perform below average at pre-test on non-verbal intelligence and/or specific reading related cognitive skills, benefit more from GraphoGame-NL exposure than children without any such risk factors.

**Question 3:** Are in-game metrics acquired from the training phase (played sessions and hours, highest game level achieved, *etc*.,) relevant predictors for the response to GraphoGame-NL intervention? *Hypothesis 3:* There is a positive relationship between exposure as measured by in-game metrics and response to intervention, and that best intervention effects are achieved by children who strictly adhere to playing GraphoGame-NL 15 min per school day for a period of seven weeks.

**Question 4:** Does an assessment of children's literacy skills by means of a dynamic assessment that is fully integrated into the game, allow us to identify the response to intervention more reliably than traditional pen-and-paper tests for word reading fluency or letter-sound knowledge? *Hypothesis 4:* Game-based assessment levels that provide online measures for response times and accuracy at the item level are more sensitive to change than traditional pen-and-paper tests as they can also capture automatisation of literacy skills.

## MATERIALS AND METHODS

For the methods and results sections, we follow the CONSORT guidelines for reporting of randomised controlled trials (*Schulz, Altman & Moher, 2010*).

### Trial design

This study employed a multicentre cluster-randomised controlled superiority trial (16 clusters across eight sites).

### Participants

Mainstream primary schools in the northern region (Groningen area) of the Netherlands and the western region (Ghent area) of the Dutch-speaking part of Belgium were contacted by phone or letter and invited to join the study. Initial requirements for participation were (a) schools had enough computers with headphones for students to play the GraphoGame-NL intervention on a daily basis, (b) classroom teachers allowed students to play the game for 10 to 15 mins per day for at least seven weeks, (c) schools allowed trained clinicians to administer behavioural tests on site at school before and after the intervention during regular school hours, and (d) teachers agreed to adhere to their allocated gaming condition. Eight schools were willing and eligible to participate (three in the Netherlands, five in Belgium) with 16 classrooms (four in the Netherlands, 12 in Belgium). All 312 children attending these classrooms were eligible to participate in the study, of which 107 lived and went to school in the Netherlands, and 205 in Belgium. Subsequently, all parents of children from the selected classrooms were informed about the study and asked for their written informed consent for the gaming and additional behavioural tests. Children were asked for oral assent prior to assessment. To enable as many children as possible to play the game, there were no initial eligibility or exclusion
criteria, and parents were also given the option to consent to participation without additional behavioural assessments. Parents (or caregivers) of participating children were asked to complete a questionnaire about their child's handedness, language(s)/dialect(s) spoken at home, family history of reading problems, neurological problems, and medication.

## Interventions

Our research group created a Dutch version of GraphoGame (GraphoGame-NL), specifically for the present study. Within the existing computerised gaming framework of GraphoGame, which was developed at the University of Jyväskylä (*Richardson & Lyytinen, 2014*), we added reading content (from letters to simple words), selected from Veilig leren lezen (VLL; 'Learning to read safely'; *Mommers, Verhoeven & Van der Linden, 1990*), a widely used literacy teaching method in both the Netherlands and the Dutch speaking part of Belgium, and a vocabulary achievement list for six-year-olds (*Schaerlaekens et al., 1999*). GraphoGame-NL included 650 items, ranging from simple and complex graphemes (*e.g.*, <n>, <r>, <ui>, over CV/VC syllables either representing separate words or occurring as parts of existing words (*e.g.*, *vi/is*), to monosyllabic words with CVC structure (*e.g.*, *vis*, 'fish') or targets with CCVC, CVCC or CCVCC consonant clusters (*e.g.*, *prijs*, 'price'; *zwart*, 'black'). For a detailed description of the tasks and materials used within the game, see Appendix 1. We excluded a few infrequent complex graphemes (<ch>, <sch>, <aai>, <auw>, <eeuw>, <ieuw>, <oei>, <ouw>) that are not typically taught at the beginning of the first grade. We also created a limited number of phonotactically legal pseudowords as minimal pairs using a pseudoword creator (Wuggy; *Keuleers & Brysbaert, 2010*).

Five female students of linguistics and speech-language pathology at the University of Groningen spoke the auditory stimuli. Native speakers of Dutch subsequently evaluated all items with respect to their prototypicality and only the most prototypical items were then included in the game, yielding one to four different spoken realisations per target. While there are some systematic differences in pronunciation between the Dutch spoken in the Netherlands and the northern part of Belgium (for phonetic distances between Dutch dialects, see *Nerbonne et al., 1996*), this should not deemed an issue as all children are exposed to standard Dutch through multimedia (movies, series, games, *etc.*).

A mathematics game specifically designed for this research was used as an active control condition. Its framework was identical to that of the reading game, featuring a range of similar reactive/interactive mini-games with varying graphics and task demands with the levels containing number/digit knowledge, counting, comparison of numbers and amounts, sorting of adjacent or nonadjacent numbers in ascending or descending order, as well as simple addition and subtraction. The range of numbers within the training goes from zero up to 20, thus mirroring the classroom content of the first half of first grade. Despite the reading and math interventions being based on the same gaming framework, there are inherent differences in cognitive load for these tasks. The math game generally features fewer distractors and shorter levels compared to the reading game, meaning that at identical exposure in terms of played sessions and hours, children playing the math game will see more levels, give more responses, but see fewer distractors on screen.

A third group of children formed a passive control group by following the conventional classroom curriculum without any additional computerised training. Notably, this group did take part in the in-game assessment sessions which the active intervention groups played in their first and last gaming session of the intervention (see details on assessments below).

The children in the two experimental groups played the respective games (reading or math) for 10 to 15 min every day during school hours, resulting in 5 to 10 min of effective task time. Children played individually on a computer or laptop wearing headphones. The supervision of the training sessions was carried out by the teachers and differed among schools depending on the numbers of computers, the curriculum, and other local circumstances. At some schools, all the children in a classroom played the respective games at the same time in a computer room, whereas children at other locations took turns using five to 10 computers during classes. To ensure that the children understood the tasks and enjoyed playing the games, at least once per week, the teachers and student assistants asked them about their progress and encouraged them to give the game another try when the content became more difficult.

## Outcomes

Pre-tests (T1) commenced in September 2015, three to six weeks after the start of the new school term, followed by a 7-week playing phase in October and November 2015, with post-testing (T2) being conducted in November and December 2015. Both behavioural assessments took place on site at schools during school hours. Testing took up to 1 h per session and was administered by undergraduate and graduate students in speech-language pathology or linguistics under the supervision of trained clinicians. Whilst the intervention groups completed the game-based assessments during the first and last playing session, the passive controls did so at some point during September or October (T1) and November or December (T2). The following outcomes were evaluated in a response to intervention paradigm with respect to the intervention efficacy of GraphoGame-NL from baseline to eight weeks follow up: reading fluency, phonological awareness, and letter-sound association.

### Reading fluency

In the pen-and-paper word reading fluency assessment at T2, students read out two custom lists of 45 words with a time limit of 1 min per list. List A contained potentially familiar or trained items (words that occurred in the game), and list B contained untrained items (words that did not occur in the game, nor in any other assessment). Words for both lists were selected from a vocabulary achievement list of 6-year-olds (*Schaerlaekens et al., 1999*) and consisted of monosyllabic words ranging from two to five letters (mean and median length of 3.5 and four letters respectively) with a frequency range of 0.3 to 36,608 per million (mean and median frequency of 1,612 and 51 per million, respectively). Based on results of 272 children, both lists correlated strongly at $r = 0.93$; split-half reliability with Spearman-Brown correction was also very high at 0.96; as was Cronbach's $\alpha$ at 0.96. Because children's performance between the lists of potentially trained and untrained

items did not differ statistically, we took the average of both lists per child and
$z$-transformed the result.

### Written lexical decision

As an in-game reading task, we implemented a written lexical decision assessment at T2
where children saw a word or pseudoword on screen and had to either accept it as a real
word or reject it as a pseudoword by clicking on a green checkbox or a red cross. This task
contained 16 words and 16 pseudowords and was split into two levels of 16 items, each
with a 3-min time limit. For data analyses, we used single-trial measures in which we
considered accuracy and response times for each word and pseudoword. Like the reading
fluency task, monosyllabic words with two to four characters (mean and median length 3.1
and three letters respectively) and a frequency range of four to 24,266 per million (mean
and median frequencies of 2,546 and 124 per million, respectively) were used.
The pseudowords were created based on those 16 words using the psycholinguistic
pseudoword generating software Wuggy (*Keuleers & Brysbaert, 2010*), in most cases, with
an edit distance of one grapheme (*e.g.*, *jas-jal* or *tijd-toed*). The pseudowords were
therefore balanced in length and featured a high neighbourhood density of real words at an
edit distance of one grapheme, ranging from 8 to 36 neighbours (with a mean and median
of 21 neighbours). Based on results of 199 children, internal consistency was questionable
as indicated by Cronbach α (0.7 and 0.68) as well as split-half reliability with
Spearman-Brown correction (0.7 and 0.65) for level and item analyses, respectively.

### Phonological awareness

Phonological awareness was assessed with two different pen-and-paper tests at T1 and T2.
First, the phonological awareness subtest of the CELF-4-NL (*Kort, Schittekatte &
Compaan, 2008*) test battery was administered, including blending phonemes into words,
identification of final and middle phonemes in words, sentence segmentation (by clapping
words), final syllable deletion, word segmentation (by clapping syllables), syllable deletion
of bi- and trisyllabic words, and initial phoneme substitution. Reliability of this test, as
measured by stratified α, is very high at 0.91 (*van den Bos & Lutje Spelberg, 2010*), as is
internal consistency measured by Cronbach's α at 0.94 (*D'hondt et al., 2008*). For analyses,
we used both $z$-transformed raw scores and norm scores, acting as both dependent and
independent variables. Second, the Proef Fonologisch Bewustzijn ('Test Phonological
Awareness', *Elen, 2006*) was presented, including rhyming, word segmentation (with the
number of syllables being indicated by clapping), blending of phonemes, syllables, or
lexemes into a word, and pseudoword repetition. No reliability measures are provided for
this test by the author.

### Timed letter-sound identification

Timed letter-sound identification was implemented into the game and assessed at both T1
and T2. Children heard a phoneme and had to select the corresponding grapheme with a
computer mouse on the screen as fast as they could. Simple and complex graphemes were
presented one by one with five to ten distractors per trial. We tested 32 different graphemes
in 42 trials distributed across four levels, each with a time limit of 1 min. The time limit

meant that only the fastest children saw all 42 targets, while slower children were only able to see a fraction of that. Based on results of 270 children, internal consistency was high as indicated by Cronbach α (0.87 and 0.93) and split-half reliability with Spearman-Brown correction (0.87 and 0.91) for level and item analyses, respectively. For data analyses, we considered both, single-trial accuracy (binary correct/incorrect) and response times as dependent variables, as well as the absolute number of correctly named letters within 4 min (letter knowledge) as a covariate. Due to their highly skewed nature, response times were always box-cox transformed (*Sakia, 1992*). For an in-depth description of the in-game assessments, see Appendix 1.

## Additional covariates

To investigate sample characteristics, the following independent variables were available from parental questionnaires: age in years, binary sex (female, male), handedness (left, right, mixed), familial risk for dyslexia (yes, no), and language spoken at home (Dutch or a foreign language). To evaluate the influence of exposure to the game, we extracted six covariates related to the individual progress children made within the game: number of sessions played, hours played, levels played, number of items seen on the screen, given responses, and maximum level achieved at the end of the intervention. For the analysis of in-game assessments, we also extracted properties of the gameplay that are not relevant for our research questions (*e.g.*, sequential trial number to adjust for autocorrelation of observations).

While there should be no unmeasured confounding in a randomised trial when randomisation was successful, adding prognostic covariates can increase power and yield more precise estimates (*Kahan et al., 2014*). We therefore measured and adjusted for abstract reasoning and rapid automatised naming in our analyses, which were not part of our research questions but are known predictors for test performance in phonological awareness (*van den Bos, 1998*) and reading fluency (*Moll et al., 2014*; *Vaessen et al., 2010*). Abstract reasoning was measured with the analogies and categories subtests of the SON-R 6-40 (*Tellegen & Laros, 2014*) as an estimate of nonverbal fluid intelligence. Reliability of this test is generally high, ranging from 0.87 to 0.95. Because norm scores are only available for children aged six and older and we had a substantial number of children under the age of six in our sample, the raw scores of both subtests were averaged and *z*-transformed. Rapid automatised naming was assessed for objects and colours (*van den Bos, 2003*). The test requires participants to name out loud 50 depicted objects and colours in five rows of ten items as accurately and as quickly as possible. We noted the time (in seconds) it took to name the entire list of 50 items. The reliability of these subtests as indicated by stratified α is in the range of 0.89 to 0.91 (*van den Bos & Lutje Spelberg, 2010*).

## Sample size

Using G*Power 3.1 (*Faul et al., 2009*) it was established that to detect medium sized group differences and/or intervention effects (Cohen's $d = 0.5$) with a power of 80% and an alpha level of 0.05 in a two-sided test, each group should contain at least 63 participants.

**Table 1 Cluster sizes and randomization.** Number of eligible children per classroom, school and country and their assignment to experimental groups.

| School | Country | Condition | | |
|---|---|---|---|---|
| | | Passive | Math | Read |
| A: 46 | **Netherlands** 107 | – | 24 | 22 |
| B: 28 | | – | – | 28 |
| C: 33 | | – | 33 | – |
| D: 71 | **Belgium** 205 | 24 | 24 | 23 |
| E: 47 | | 11 | 18 | 18 |
| F: 49 | | 17 | 16 | 16 |
| G: 28 | | – | 14 | 14 |
| H: 10 | | 10 | – | – |
| **Total** | 312 | 62 | 129 | 121 |

## Randomisation

Since it was deemed too difficult and logistically challenging for teachers to ensure that every child played according to an individually randomised gaming condition, we used clustered randomisation to assign the gaming condition by classroom. Therefore, 16 clusters, each containing 10 to 33 children, were semi-randomly assigned to either play the reading version of GraphoGame-NL, a math version of GraphoGame (active controls), or attend the normal school curriculum (passive controls). Where possible, a within-school design was set up: three schools participated with three classrooms, so each of the three gaming conditions was randomly assigned to one of the three classrooms. Another two schools joined with two classrooms, where the reading and math game were randomly assigned to each classroom. The final three schools joined with one classroom and therefore, each school was assigned to one of the three gaming conditions. See Table 1 for an overview of the number of eligible children within each cluster and their assigned gaming conditions.

## Blinding

Children and teachers had to be aware of the gaming condition they were assigned to and could not be blinded in our design. During the statistical analysis, those assessing the outcomes were also not blinded to group allocation.

## Statistical methods

Statistical analyses were conducted in R (Version 4.1, *R Core Team, 2021*). Differences in baseline measures between the gaming conditions (reading, math, passive control) and countries (Netherlands, Belgium) were tested using two-way ANOVAs. Significant main effects or interactions were then followed up with *t*-tests or Tukey HSD tests.

The evaluation of intervention effects was conducted with linear mixed effects regression (*Bates et al., 2015*). One mixed regression model was fit for each of the seven outcome variables at T2: word reading fluency, written lexical decision (accuracy and response

time), phonological awareness (CELF and Proef), and timed letter-sound identification (accuracy and response time). To facilitate interpretation, the outcomes were centred and z-transformed where possible, so that the model coefficient $\beta$ is identical to the effect size Cohen's d (*Baguley, 2009*).

Due to the large number of potential covariates and the explorative nature of the research questions two through four, it was not feasible to set up hypothesis driven models as these do not converge with the given sample size. We therefore opted to use a data driven approach and identify the best fitting model based on Akaike's Information Criterion which penalises additional covariates Akaike's Information Criterion (AIC; *Akaike, 1974*). We tested the stepwise forward inclusion of main effects and interactions of all covariates mentioned above (fitted with maximum likelihood), as well as random intercepts and slopes of subjects, classrooms, and schools for the random effects structure (fitted with restricted maximum likelihood estimation). Each coefficient was only kept if it reduced AIC by at least two, resulting in a model which is at least 2.7 times as likely regarding the evidence ratio (*Anderson & Burnham, 2004*). Where available, we always tested for inclusion of raw and percentile/norm scores as predictor (*e.g.*, CELF phonological awareness yielded both raw and norm scores, we tested the inclusion of both, and if they reduced AIC by at least two, we picked the one with the lower AIC).

To ensure that presented effects are not carried by outliers, for each resulting model, we trimmed observations based on residuals beyond ±2 standard deviations of the model prediction and refitted the model. Each model then underwent model criticism to ensure that reported models fulfil regression assumptions of independence of observations as well as a normal and homoscedastic distribution of residuals. Usually, model fit is evaluated by the squared correlation between the observed and the fitted values ($R^2$). For mixed-effects models, this method can only estimate the residual variance and thus ignores the random effects present in the model. Following the approach proposed by *Nakagawa & Schielzeth (2013)*, marginal and conditional $R^2$ were calculated, instead. The former is an estimation of the fixed-effects structure alone, while the latter incorporates both fixed and random effects.

As several of the fitted models did not show intervention effects, the Results section will focus on those models that show effects related to the gaming conditions and our research aims (see Table 2 for specifications of all the fitted models for each outcome and the supplementary R markdown for all results).

## Changes to statistical methods because of baseline analyses

Baseline comparisons showed main effects of country (see Results of Baseline data below), whereby the children in the Dutch sample consistently outperformed the Belgian children in timed letter-sound knowledge, phonological awareness, and rapid automatised naming at T1. Therefore, contrary to the initial analysis plan, the Belgian and the Dutch samples were used independently to evaluate all research questions. This doubled the number of planned analyses from 7 to 14 models and reduced statistical power due to smaller group sizes.

In addition, the described approach did not yield answers for research question three in that none of the game exposure measures ended up as relevant predictors (see Results of

**Table 2 Overview of fitted models and results.** Overview of the 15 models that were fitted to answer the research questions. For each of the seven outcome measures two models were fitted: one for each country. For word reading fluency, an additional model combining the two countries was used. Most models describe null results, so the results section focusses on those models that show an effect of experimental condition.

| Outcome variable at T2 | Country | N | $R^2$ | Effect of condition | Included relevant co-variates (according to AIC) | | | | | | |
|---|---|---|---|---|---|---|---|---|---|---|---|
| | | | | | Age | Sex | FR | T1 LK | T1 PA | T1 RAN | T1 IQ |
| Word reading fluency | B | 150 | 0.39 | Read > Passive | | | | ✓ | ✓ | ✓ | |
| CELF PA | | 152 | 0.64 | Math > Passive | | | | | ✓ | | ✓ |
| PROEF PA | | 150 | 0.46 | n.s. | | | ✓ | | ✓ | | |
| LSSI accuracy | | 104 | 0.47 | Read > Math > Passive | | | | | ✓ | | |
| LSSI speed | | 108 | NA | n.s. | | | | ✓ | | ✓ | |
| WLD accuracy | | 104 | 0.16 | n.s. | | | | | ✓ | | |
| WLD speed | | 103 | 0.42 | n.s. | ✓ | | | ✓ | ✓ | | |
| Word reading fluency | NL | 78 | 0.43 | n.s. | | | | ✓ | ✓ | | |
| CELF PA | | 83 | 0.45 | n.s. | | | | ✓ | ✓ | | |
| PROEF PA | | 81 | 0.52 | n.s. | | | | | ✓ | | |
| LSSI accuracy | | 75 | 0.81 | n.s. | | ✓ | | | ✓ | | |
| LSSI speed | | 75 | NA | Read > Math | ✓ | | | | ✓ | | |
| WLD accuracy | | 75 | 0.65 | n.s. | | | | | | | |
| WLD speed | | 75 | 0.54 | n.s. | | | | ✓ | ✓ | | |
| Word reading fluency | B + NL | 196 | 0.49 | Read > Math | | | | | ✓ | ✓ | |

**Note:**
NL, Netherlands; B, Belgium; PA, phonological awareness; LSSI, timed letter-sound identification; WLD, written lexical decision; LK, letter knowledge; RAN, rapid automatized naming; IQ, abstract reasoning; NA, not available ($R^2$ is not computable for some LSSI response time models due to presence of random slopes); T1, pre-test; n.s., not significant; tick marks indicate co-variates included in the best model as chosen by AIC.

Question 3: Intervention properties below). In an explorative approach, we therefore re-combined children from both countries to increase statistical power, merged all exposure measures by means of a principal component analysis and fitted two additional models for the outcome of reading fluency using non-linear mixed effects regression (Generalized Additive Model; *Wood, 2006*).

### Research ethics

This research was approved by the ethical committee of the Faculty of Arts of the University of Groningen, and the Faculty of Psychology of the University of Ghent (2015/25).

## RESULTS

### Participant flow

Out of 312 children in the selected classrooms, parents of 26 children did not give consent to participate. Eight children were lost to follow up as they did not attend the second assessment. For the final analysis, we also excluded data of children that were allowed to play but did not consent to the behavioural assessments ($N = 4$), were repeating the first grade ($N = 2$), were one year older than their peers without repeating first grade ($N = 1$), and those who were diagnosed with a neurodevelopmental disorder ($N = 5$). To conduct a

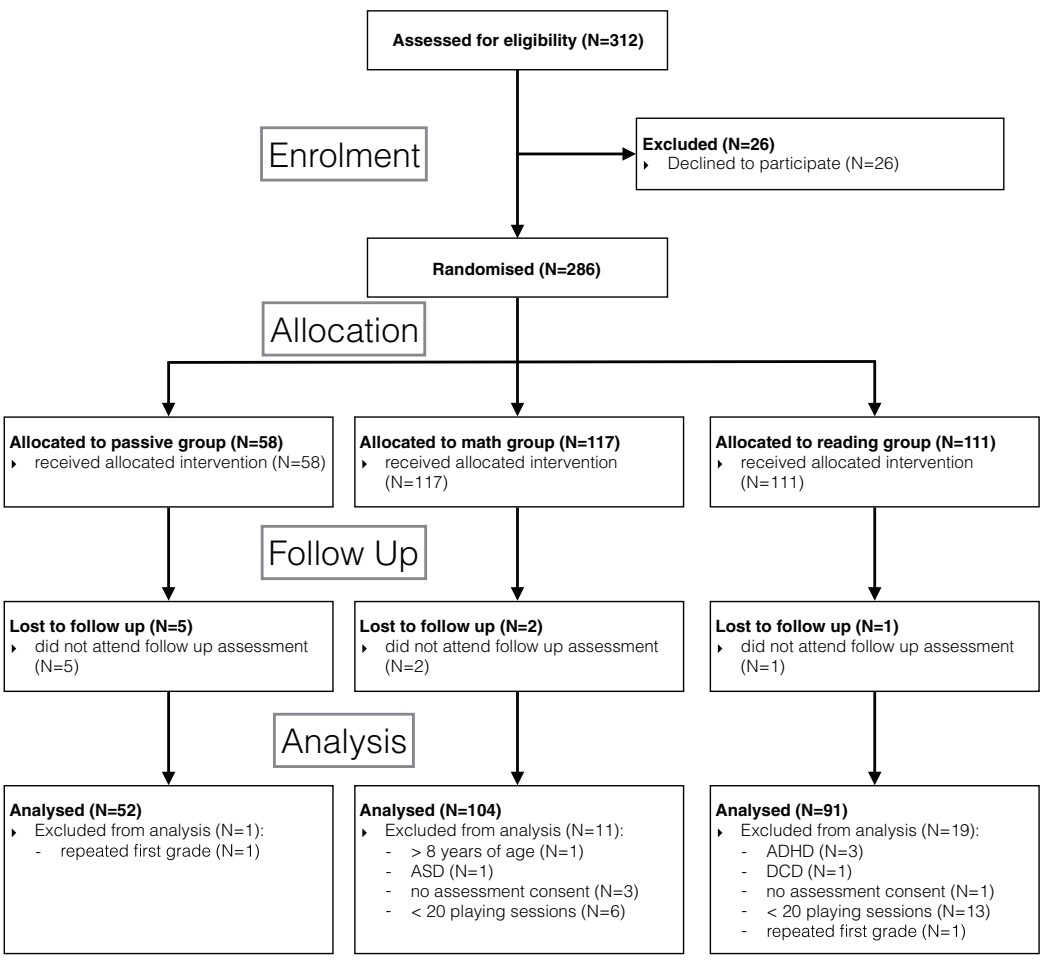

**Figure 1 Flow diagram.** Consolidated Standards of Reporting Trials (CONSORT) flow diagram showing the cluster randomized trial at the subject level.

response to intervention analysis, we further excluded children who failed to play at least 20 sessions (corresponding to four weeks of daily playing) or alternatively failed to accumulate at least 2.5 h of game exposure ($N = 19$). The details of participant flow are specified in the CONSORT flow diagram (Fig. 1).

## Data losses and exclusions

Due to cases of missing observations and trimming of outliers after model fitting, most analyses were conducted on smaller subsets of the data. Exact sample sizes are reported at the corresponding positions of the results section. Most notably, due to data retrieval problems, the results of the in-game assessments at T2 were lost for 66 children putting the available sample size at 60, 71, and 49 participants for the reading, math, and passive group, respectively.

## Baseline data

At T1, the Dutch sample significantly outperformed its Belgian peers in terms of abstract reasoning ($F_{1,242} = 15.74$, $p < 0.001$), letter knowledge ($F_{1,241} = 288.84$, $p < 0.001$), both

**Table 3 Participant characteristics.** Descriptive statistics of the three experimental groups by country at T1 and T2 ($N = 247$). Values represent counts (percentages) or means (standard deviations).

| | Passive B | Passive NL | Math B | Math NL | Read B |
|---|---|---|---|---|---|
| **Country** | B | NL | B | NL | B |
| $N$ | 52 | 48 | 56 | 38 | 53 |
| **Sex** (f/m) | 21/31 *(40%)* | 23/25 *(48%)* | 21/35 *(38%)* | 23/15 *(61%)* | 24/29 *(45%)* |
| **Age** (years) | 6.20 *(0.30)* | 6.25 *(0.32)* | 6.31 *(0.31)* | 6.19 *(0.29)* | 6.26 *(0.37)* |
| **Familial risk** (yes/no) | 7/45 *(14%)* | 11/37 *(23%)* | 7/49 *(13%)* | 6/32 *(16%)* | 10/43 *(19%)* |
| **Handedness** (l/r) | 9/43 *(17%)* | 3/45 *(6%)* | 5/51 *(9%)* | 6/32 *(16%)* | 4/49 *(8%)* |
| **Monolingual** (y/n) | 46/6 *(88%)* | 36/12△ *(75%)* | 53/3 *(95%)* | 36/2△ *(95%)* | 51/2 *(96%)* |
| **Abstract reasoning**[†] (z-score) | −0.29 *(0.99)* | 0.41 *(0.81)* | −0.15 *(1.02)* | 0.48 *(1.07)* | −0.06 *(0.88)* |

| Session | Passive B T1 | Passive B T2 | Passive NL T1 | Passive NL T2 | Math B T1 | Math B T2 | Math NL T1 | Math NL T2 | Read B T1 | Read B T2 |
|---|---|---|---|---|---|---|---|---|---|---|
| **Letter knowledge**[†] | 11.27△ *(5.30)* | 21.32 *(4.92)* | 24.50 *(4.74)* | 28.46 *(3.45)* | 9.36 *(6.42)* | 20.78 *(5.97)* | 22.24 *(6.36)* | 25.26 *(6.11)* | 8.23△ *(6.70)* | 22.58 *(6.84)* |
| **CELF PA**[†] (percentile) | 41.14 *(23.91)* | 50.50 *(24.37)* | 59.85 *(19.35)* | 72.50 *(19.26)* | 38.22 *(23.68)* | 55.71 *(25.32)* | 63.68 *(22.22)* | 73.46 *(17.63)* | 35.74 *(20.05)* | 50.87 *(20.49)* |
| **PROEF PA**[†] (percentile) | 49.52 *(25.94)* | 55.91 *(29.02)* | 64.84 *(26.55)* | 73.80 *(19.63)* | 45.67 *(29.28)* | 61.03 *(26.92)* | 72.50 *(22.80)* | 75.92 *(22.75)* | 44.29 *(25.37)* | 48.73 *(24.33)* |
| **RAN colours**[†] (seconds) | 70.88 *(15.92)* | 57.37 *(12.57)* | 58.44 *(11.80)* | 52.46 *(10.17)* | 67.50△ *(15.37)* | 55.88 *(11.42)* | 59.39 *(14.71)* | 53.11 *(11.87)* | 74.96△ *(16.02)* | 59.40 *(14.01)* |
| **RAN objects**[†] (seconds) | 75.56 *(17.49)* | 68.08 *(19.37)* | 70.46 *(15.11)* | 66.54 *(17.76)* | 76.25 *(17.33)* | 69.93 *(21.50)* | 65.68 *(12.13)* | 63.67 *(15.70)* | 81.77 *(23.82)* | 73.55 *(22.18)* |
| **Word reading** (words per minute) | – | 10.05 *(4.59)* | – | 23.83 *(11.67)* | – | 12.02 *(7.79)* | – | 19.34 *(10.29)* | – | 12.62 *(7.43)* |

**Notes:**
[†] Significant difference between countries at T1 ($p < 0.05$).
△ Significant difference between conditions within countries at T1 ($p < 0.05$).
NL, Netherlands; B, Belgium; PA, phonological awareness; RAN, rapid automatized naming; T1, pre-test; T2, post-test.
Italicized values refer to percentages and standard deviations.

phonological awareness tests (CELF: $F_{1,238} = 59.68$, $p < 0.001$; PROEF: $F_{1,242} = 37.81$, $p < 0.001$), and both rapid automatised naming measures (colours: $F_{1,242} = 31.40$, $p < 0.001$; objects: $F_{1,241} = 16.58$, $p < 0.001$; see Table 3). For this reason, separate analyses were carried out for the Dutch and the Belgian sample, as referred to also in the Statistical Methods section, above. For the Belgian sample, there was a main effect of condition for rapid automatised naming of colours at T1 ($F_{2,158} = 3.06$, $p = 0.050$), where the math group was significantly faster than the reading group (*post-hoc* Tukey HSD test: $p = 0.039$). There was also a main effect of condition for letter knowledge ($F_{2,157} = 3.22$, $p < 0.043$), where the passive group knew more letters than the reading group (*post-hoc* Tukey HSD test: $p = 0.035$). For the Dutch sample, the math group had significantly more children who did not speak Dutch at home than the reading group (Fisher's exact test: $p = 0.018$).

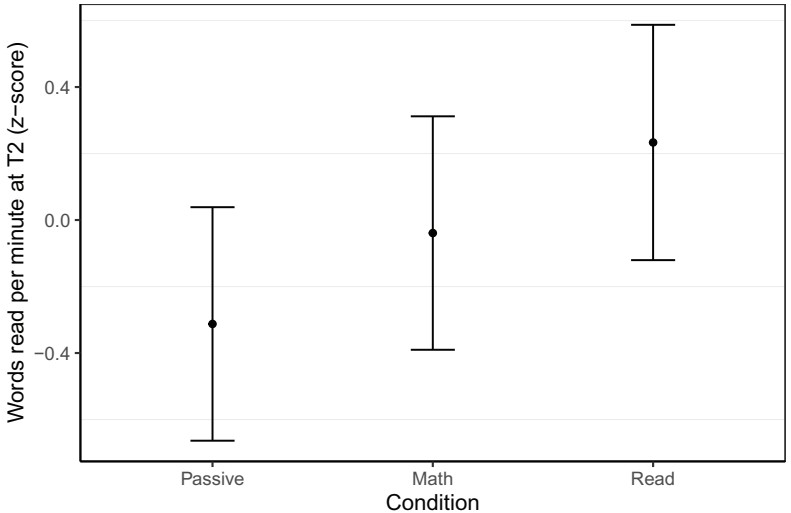

**Figure 2  Word reading fluency within the Belgian sample at T2.** Model predicted z-transformed 1 min word reading fluency scores of the Belgian sample at T2. Whiskers represent 95% confidence intervals.

## Question 1: response to intervention

To answer the first research question whether children playing GraphoGame-NL show a larger response to intervention than children playing a control game or not playing at all, we assessed word reading fluency and phonological awareness with pen-and-paper tests. We also used two in-game tests to assess both response times and accuracy in written lexical decision and timed letter-sound identification tasks.

### Word reading fluency

Word reading fluency was assessed with two, 1-min reading lists at T2 (see Table 3). While we did not find any effects associated with gaming condition in the Dutch sample, there were effects in the Belgian group, as shown in Fig. 2. At T2, neither the reading ($\beta = 0.27$, $t = 1.60$, $p = 0.09$) nor the passive ($\beta = -0.27$, $t = -1.63$, $p = 0.106$) group differed from the math group, but the reading group outperformed the passive group ($\beta = 0.55$, $t = 3.18$, $p = 0.002$). In terms of effect sizes, these differences were small ($d = 0.27$) for the passive and reading group compared to the math group, and medium-sized ($d = 0.55$) when comparing the reading group to the passive group. This best model was based on 150 children ($N_{\text{Passive}} = 48$, $N_{\text{Math}} = 52$, $N_{\text{Read}} = 50$, trimmed seven observations or 4.3% of data), controlled for letter knowledge at T1, CELF phonological awareness at T1, log transformed rapid automatised naming of colours time at T1, and included random intercepts per school. The model had a conditional $R^2$ of 0.39 and a marginal $R^2$ of 0.30.

### Written lexical decision

The written lexical decision assessment as an additional measure of reading abilities was embedded into the last gaming session of the intervention at T2 and did not reveal any differences between groups in terms of accuracy or response times.

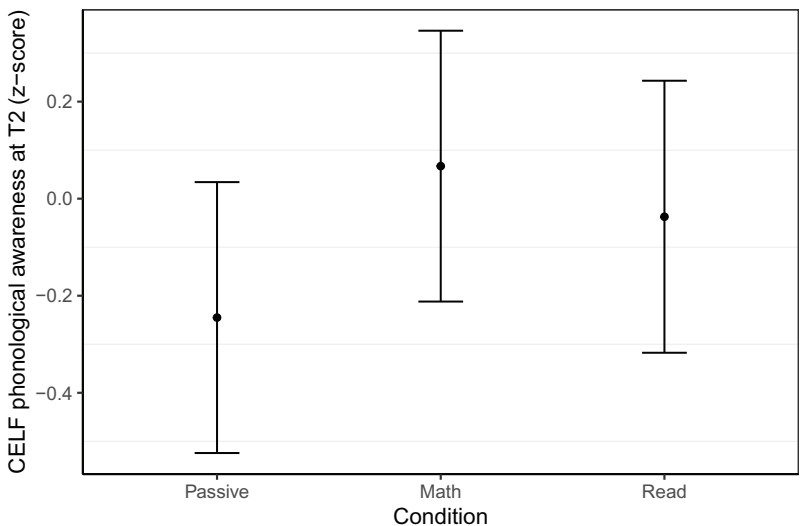

**Figure 3 Phonological awareness within the Belgian sample at T2.** Model predicted z-transformed phonological awareness scores as measured with the CELF-IV NL of the Belgian sample at T2. Whiskers represent 95% confidence intervals.

### Phonological awareness

Phonological awareness at T2 was measured using the nine subtests of the CELF-IV and four subtests of the Proef. Again, we did not find any effects related to gaming condition in the more advanced Dutch sample, but found an effect for the CELF-IV in the Belgian sample (see Fig. 3). The math group outperformed the passive group ($\beta$ = 0.31, $t$ = 2.36, $p$ = 0.020) but did not differ from the reading group ($\beta$ = 0.10, $t$ = 0.83, $p$ = 0.407), nor did the reading group differ from the passive group ($\beta$ = 0.21, $t$ = 1.55, $p$ = 0.123). This best model was based on 152 children ($N_{Passive}$ = 48, $N_{Math}$ = 52, $N_{Read}$ = 52, trimmed five observations or 3.1% of data), controlled for abstract reasoning at T1, CELF phonological awareness at T1, Proef phonological awareness at T1 and included random intercepts per school. The model had a conditional $R^2$ of 0.64 and a marginal $R^2$ of 0.54.

### Timed letter-sound identification

The timed letter-sound identification assessment was embedded into the game itself, taking place in the first and the last gaming session of the intervention for T1 and T2 respectively. From T1 to T2, the reading game boosted accuracy in this task for the Belgian sample and we saw a trend towards faster response speed in the Dutch sample. For the Belgian sample, the best model predicting single-trial accuracy at both testing sessions for 6756 trials of 101 children ($N_{Passive}$ = 47, $N_{Math}$ = 21, $N_{Read}$ = 33), revealed an interaction of time × condition (see Fig. 4). At T1, the passive control group knew significantly more letters than the math and reading groups (math: $\beta$ = 0.53, $z$ = 2.37, $p$ = 0.018, reading: $\beta$ = 0.51, $z$ = 2.60, $p$ = 0.009). While we did find a main effect of time for the math group ($\beta$ = 1.78, $z$ = 12.69, $p$ < 0.001), this was significantly smaller for the passive group ($\beta$ = −0.49, $z$ = −3.02, $p$ = 0.003) and somewhat bigger for the reading group ($\beta$ = 0.34, $z$ = 1.88, $p$ = 0.060). The gain in accuracy of the reading group far exceeded that of the passive group ($\beta$ = 0.83, $z$ = 5.79, $p$ < 0.001). This best fitting model controlled for game

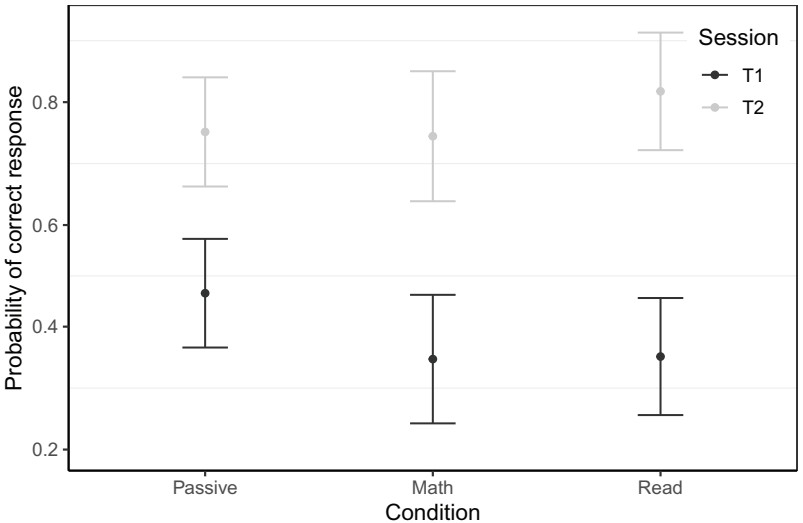

**Figure 4 Timed letter-sound identification accuracy scores by session within the Belgian sample.**
Model predicted accuracy for the in-game letter-sound identification task of the Belgian sample.
Whiskers represent 95% confidence intervals. 

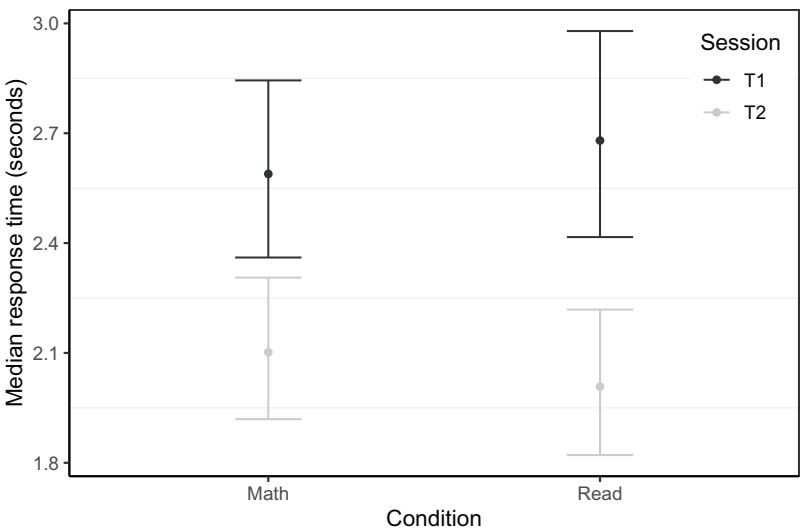

**Figure 5 Timed letter-sound identification response times by session within the Dutch sample.**
Model predicted response times for the in-game letter-sound identification task of the Dutch sample.
Whiskers represent 95% confidence intervals. 

level, CELF phonological awareness at T1 and response time of the current trial.
The random effect structure consisted of random intercepts per subject and target, and random slopes for previous trial response time and CELF phonological awareness at T1 by subject. The model had a conditional $R^2$ of 0.47 and a marginal $R^2$ of 0.20.

For the Dutch sample, the best model, which predicted box-cox transformed single-trial response times based on 3646 trials of 75 children ($N_{Math} = 48$, $N_{Read} = 27$, trimmed 212 trials or 4.9% of data), also revealed a time × condition interaction (see Fig. 5). At T1, the reading and the math groups did not differ ($\beta = 0.01$, $t = 0.59$, $p = 0.597$), and we found a

significant main effect of time ($\beta = 0.04$, $t = 7.76$, $p < 0.001$). In addition, a marginally significant interaction of group and time ($\beta = 0.01$, $t = 1.98$, $p = 0.052$) indicates that the speed increase from T1 to T2 was bigger for the reading than the math group. This best model controlled for PROEF phonological awareness at T1, age at T1, trial number, and previous trial response time. The random effects structure consisted of intercepts per subject, class, target, and distractor order on screen, as well as random slopes for time by subject and random slopes for time by target.

## Question 2: participant characteristics

To answer the second research question whether certain subgroups of children benefit more from GraphoGame-NL exposure than others, we looked for possible interaction effects of gaming condition with pre-test scores, age, binary sex, familial risk for dyslexia, abstract reasoning, and home language environment, on the above-presented response-to-intervention variables. In almost all analyses, participant characteristics explained unique variance as covariates and thus helped to describe more robust and generalisable intervention effects. However, we did not find statistically significant interaction effects between participant characteristics and gaming condition, indicating that there were no participant characteristics that modulated response specifically to GraphoGame-NL intervention.

Familial risk for dyslexia was assessed by parental questionnaires inquiring about the occurrence of reading difficulties in first degree relatives. According to the stepwise model building, familial risk was a relevant predictor for PROEF phonological awareness scores at T2 in the Belgian sample, reflected in slightly lower scores for children with a familial risk for dyslexia across all gaming conditions ($\beta = -0.23$, $t = -1.34$, $p = 0.183$) with a small effect size ($d = 0.23$). Otherwise, we found no evidence that familial risk for dyslexia affected response to intervention.

Age was a relevant covariate in analyses of in-game response times of timed letter-sound identification ($\beta = 0.02$, $t = 2.22$, $p = 0.030$) at both testing points in the Dutch sample and of response times in the written lexical decision tasks ($\beta = -0.20$, $t = -3.17$, $p = 0.002$) at T2 in the Belgian sample. In both cases, on average, younger children took longer to respond than older children.

Sex was a relevant covariate for letter-sound identification accuracy in the Dutch sample at both testing points ($\beta = -1.62$, $t = -4.59$, $p < 0.001$) and for word reading fluency at T2 in the combined sample ($F = 8.78$, $p < 0.001$). In both cases, girls outperformed their male peers by a significant margin.

Nonverbal intelligence, as measured by the SON-R 6-40, was a relevant covariate for the CELF phonological awareness at T2 in the Belgian sample ($\beta = 0.17$, $t = 2.65$, $p = 0.009$), with higher nonverbal intelligence at T1 associating with higher phonological awareness scores at T2 ($d = 0.17$).

Home language environment and handedness were never relevant predictors.

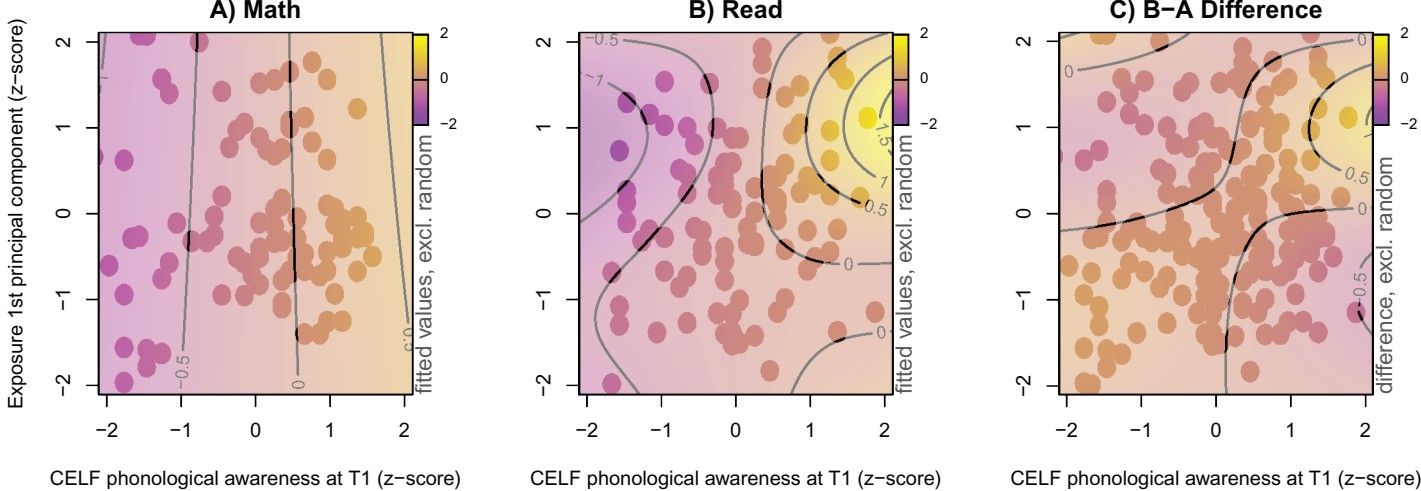

**Figure 6 Word reading fluency by experimental group within the combined sample at T2.** Model predicted 1 min word reading fluency at T2 (colour coded) of the two experimental groups after combining both countries using a nonlinear game exposure × CELF PA at T1 interaction. All variables are z-transformed. Dots indicate available observations. (A) Math group. (B) Reading group. (C) Difference of both groups.

## Question 3: intervention properties

To answer the third research question, whether in-game metrics acquired from the training phase would be relevant predictors for training outcomes, we looked for possible associations between game exposure and response to intervention. Exposure measures potentially reflecting learning opportunity were the number of played sessions, played hours, played levels, seen items, given responses and the maximum game level reached by the end of the intervention. The separate inclusion of these six measures of game exposure and progress was tested in all models fitted for the seven outcomes mentioned above and in no case were they relevant predictor for intervention outcomes (see Results of Question 1: Response to intervention).

As an additional explorative analysis, we focussed on the word reading fluency outcome, re-included children who played less than 20 sessions ($N = 19$), and merged children from both countries to increase statistical power ($N = 210$). For this purpose, one additional model was fitted, in which the word reading fluency score was modelled nonlinearly as a function of an interaction of continuous independent variables (principal component of the six game exposure variables, phonological awareness, rapid automatised naming, letter-sound knowledge). The best model contained two nonlinear interaction smooth surfaces of CELF phonological awareness at T1 and the first principal component of the six exposure variables. For the reading group, this nonlinear interaction was significant ($F = 2.99$, $p = 0.009$), while for the math group, it was not ($F = 0.20$, $p = .653$). As shown in Fig. 6, within the math group (subplot A), there was an almost linear relation between phonological awareness skills at T1 and reading fluency as indicated by the vertical and equidistant topographic lines, whereas, for the reading group (subplot B), there was a nonlinear interaction of these two variables. The effect of phonological awareness on the

reading fluency outcome was positive when exposure was above average (*i.e.*, a *z*-score around 1), and absent when exposure was below average (*i.e.*, a *z*-score around −1).

## Question 4: assessment tools

Our fourth research question asked whether an assessment of literacy skills by means of a dynamic assessment that is fully integrated into the game, allows us to identify the response to intervention more reliably than traditional pen-and-paper tests. For the assessment of reading fluency, we found intervention effects and group differences with a conventional 1-min reading, pen-and-paper test (see Results of Word reading fluency), which the in-game assessment did not capture (see Results of Written lexical decision). In contrast, we were able to show the advantage of obtaining in-game data of letter-sound knowledge at the item level. Analysing single-trial data showed that children who played the literacy game made more pronounced progress than their peers who played the math game or who did not play any game (see Results of Timed letter-sound identification), whereas analysing aggregated data of the same task (*i.e.*, by creating a count of correctly named letters within 4 min) with an ANOVA, did not reveal the same effects in terms of group differences and interactions (all $p > 0.1$).

## DISCUSSION

In this study, we evaluated the effectiveness of a newly created version of GraphoGame for Dutch-speaking beginning readers, employing active (math game) and passive (no game) control conditions in 16 first-grade classrooms in the Netherlands and Flanders. The main purpose of this game was to intensify exposure to relevant early reading materials and to provide additional training for struggling beginning readers on top of mainstream reading instruction in the classroom. The novelty in our study was not only a newly created Dutch adaptation of an existing game-based literacy framework, but also the inclusion of entire first-grade classrooms, in-game assessments and exposure parameters yielding individualised metrics allowing accurate monitoring of individual progress throughout the game. Beyond an overall evaluation of response to intervention, we explored three factors to determine the effectiveness of digital game-based learning in early literacy training within the framework of a single study: participant characteristics, intervention properties, and assessment tools.

## Question 1: response to intervention

For our first and main research question, we wanted to evaluate the effect of playing GraphoGame-NL for up to seven weeks at the onset of formal reading instruction, and hypothesised that children playing GraphoGame-NL would show a larger response to intervention compared to children playing a control game or children who did not play at all. This hypothesis was partly confirmed. In the Belgian sample, children who played the literacy game improved their letter-sound knowledge more than the math and passive control groups (as measured by the accuracy in the timed letter-sound identification task). In addition, we observed faster word reading fluency in this group at T2 with small to medium-sized effects compared to the children assigned to the passive control and, to a

lesser extent, the math condition. For the Dutch sample, there was a trend towards faster responses in the timed letter-sound identification task for the reading group compared to the math group. Recombining both samples revealed a nonlinear interaction of exposure to the game and phonological awareness scores at T1 for word reading fluency. Children who scored high on phonological awareness prior to training and played extensively were more fluent readers than could be expected based on phonological awareness and rapid automatised naming alone.

## Question 2: participant characteristics

As for the second research question about participant characteristics, we asked whether there are certain subgroups of children who benefit more from GraphoGame-NL exposure than others and hypothesised that poor performers would benefit more. However, the effects relating to gaming condition were mostly main effects, indicating that there were no systematic differences between participants' proficiency levels across the three experimental groups. The only exception, pointing in the opposite direction to our hypothesis, was that few children who performed above average in phonological awareness skills at pre-test who were comparatively faster readers when they had above average exposure to the reading game. We also anticipated that certain subgroups of children, like those at familial risk or those speaking a different language at home, might perform worse at pre-test and exhibit a different outcome from exposure to the game, but this was not evident in our findings.

Most studies use an inclusion criterion based on scores in reading-related tests (*e.g.*, *Saine et al., 2010*, *2011*), the nomination by class teachers (*e.g.*, *Kyle et al., 2013*), or socioeconomic status (SES, BetzdoBetzdorf; *e.g.*, *Rosas et al., 2017*). While the rationale for such inclusion criteria is clear, all these approaches pose certain difficulties. In the case of the test-based or SES-based approach, there is the question of finding the right cut-off score. Children scoring at the lower end of the population scale are more likely to perform closer to average at the next assessment, a phenomenon known as regression to the mean (*Morton & Torgerson, 2003*). Furthermore, teacher ratings may be subjective and based on the assessment of skills unrelated to a child's reading abilities (*Begeny et al., 2011*).

To prevent such sampling bias in the present study, we invited all children from 16 classrooms to play, independent of their performance on reading-related tasks and investigated the effect of pre-test scores on training-induced skill improvement. Our approach was unintentionally strengthened further because of the large pre-test differences between the Dutch and Belgian children in our sample. These differences appear to stem from the different preschool systems, where Belgium has a stricter separation of pre-school and school, compared to the more gradual transition into formal instruction from four years of age onwards in the Netherlands. Similar differences between these two neighbouring countries have been observed in early numeracy skills (*Torbeyns et al., 2002*). Ultimately, this gave even further spread to the preliteracy skills in our sample and allowed us to evaluate the impact of factors such as age, familial risk for dyslexia, sex, home language environment, handedness, and an intelligence measure (*i.e.*, abstract reasoning) more exhaustively than previous research.

At first sight, one could argue that, due to the absence of interactions of pre-test scores and outcome, the intervention was equally effective for all children. However, when comparing results stratified by country, it is apparent that the weaker beginning readers in Belgium showed more intervention effects (both in letter-sound knowledge and word reading fluency), whereas in the more advanced Dutch sample, we found fewer effects (limited to grapheme-phoneme correspondence automation). This can be taken as evidence that individual starting levels matter for GraphoGame-NL intervention outcomes, in line with most previous studies. Training poor performers at an early stage in their literacy development usually yields group-wide benefits in easily trainable skills like letter knowledge (*e.g.*, *Brem et al., 2010*; *Rosas et al., 2017*), and, in longer interventions, also decoding and reading (*e.g.*, *Saine et al., 2010*, *2011*). However, the opposite effect, that children with high pre-test scores have an increased benefit, has also been reported. *Ruiz et al. (2017)* found a small but significant advantage for early readers who already scored high at pre-test in timed letter knowledge. The few studies that trained entire classrooms (*e.g.*, *Jere-Folotiya et al., 2014*; *Koikkalainen, 2015*; *Ronimus & Lyytinen, 2015*) unfortunately did not consider interaction terms with pre-test scores in their analyses, thus provide no reference point for comparisons. Regarding the general role of pre-test scores as predictors for intervention outcomes, conventional reading interventions found that reading-related skills are poor predictors for the response to intervention. Improvements were rather related to levels of short-term memory and vocabulary (*Byrne, Shankweiler & Hine, 2008*)—two variables which were not measured in the present study and are not routinely collected and used as covariates in analyses of reading interventions.

### Familial risk of dyslexia

For effects relating to familial risk of dyslexia, we found that at-risk children score slightly lower across both assessment points only with respect to phonological skills, and that status of familial risk did not influence the training effectiveness. The former is somewhat surprising, given that other studies also reported weaker performance in other reading precursors for children at familial risk like rapid automatised naming (*van Bergen et al., 2012*; *Lyytinen et al., 2004*). So far, only two studies have specifically investigated the role of familial risk in GraphoGame effectiveness. While a study by *Brem et al. (2010)* did not find any distinct effects relating to familial risk either, a study by *Blomert & Willems (2010)* found that at-risk children did not improve as much as peers. The authors concluded that familial risk of dyslexia is characterised by a letter-sound association and integration deficit, which the data from the present study does not support. The fact that the present study did not find distinct training effects attributable to familial risk may be due to the small number of at-risk children in each condition (varying from seven to 18) or the rather weak self-report questionnaire asking for reading failure in the close family, but without requesting proof of a formal diagnosis in first degree relatives.

### Sex

In our sample, boys had significantly poorer letter-sound knowledge and phonological awareness skills compared to girls at the start of first grade. This appears to be the onset of

a constant difference which extends throughout school into adolescence, where girls outperform boys in terms of reading (*OECD, 2010*; *Ming Chui & McBride-Chang, 2006*; *Tops et al., 2019*; *Torppa et al., 2015*). Sex differences therefore warrant scrutiny in literacy digital game-based learning research, also given that boys generally play more games and show a stronger preference for game-based learning than their female peers (*Nietfeld, Shores & Hoffmann, 2014*; *Admiraal et al., 2014*; *Gwee, San Chee & Tan, 2011*; *Bonanno & Kommers, 2007*). Ideally, studies should therefore control for sex or previous game experience in their analyses, which is currently almost never done in the field *e.g.*, for studies reported in *McTigue et al. (2020)*.

## Question 3: intervention properties

Concerning the third research question of intervention properties, we asked whether in-game metrics are relevant predictors for response to GraphoGame-NL intervention. Studies reporting positive GraphoGame-related effects used training durations ranging from one up to 28 weeks with an intensity of two to five training sessions per week (*McTigue et al., 2020*; *Richardson & Lyytinen, 2014*). However, whether training duration and intensity act as independent variables modifying digital game-based learning outcomes, or whether the overall exposure to the game (in hours) is a better predictor of training effectiveness, remain open questions. Furthermore, the ideal training duration and intensity may depend on population properties and training goals, which raises the obligation to investigate possible interactions of training and population properties.

Previous literacy digital game-based learning studies using GraphoGame usually rely exclusively on the number of gaming sessions, or the time spent playing as a measure of training intensity. Only a few studies communicate treatment fidelity measures such as attrition rates, which can be as high as 46% (*Jere-Folotiya et al., 2014*). We therefore extracted additional game-exposure measures, such as the highest level attained, or total number of seen items which might capture the actual gameplay better than mere task time. For example, even though all children played in the range of 20–30 sessions, the number of items seen within the training period had a much wider range from 5,000 to 20,000. This is a result of the speed and accuracy of children: responding faster results in more levels, responses and seen items, while being less accurate results in being exposed to fewer items during the same task time. Due to the adaptivity of modern games which constantly adjust the difficulty level to the individual learner, different children are exposed to different content, making exposure comparisons difficult, even within the same study. Response patterns also vary over time depending on the complexity (simpler, more familiar content *vs.* more complex new information) of consecutive levels (*Njå, 2019*). We therefore hypothesised that characteristics from the gaming process itself might help explain variance in the intervention outcome. Our study provides some evidence in this direction, in that exposure to the game was positively related to reading fluency outcome, although this only applied to the participants with above average phonological awareness skills. This suggests that data extracted from in-game behaviour can indeed be used for dynamic individual assessment which implies that GraphoGame-NL can serve a diagnostic function

by identifying non-responders at an early stage of literacy development (*Koikkalainen, 2015*; *Puolakanaho & Latvala, 2017*).

Possibly, the rather strict inclusion criterion of at least 20 playing sessions made the present sample too homogenous to find interactions between intervention outcomes and exposure. Upon re-inclusion of children who played less than 20 sessions and by combining exposure measures with a principal component analysis, we found that game exposure modulated reading fluency when phonological awareness and rapid automatised naming were statistically controlled. For the maturation of literacy skills, we argue that the time-course of development of phonological skills plays a crucial role for the benefits of GraphoGame-NL. Playing beyond mastery of grapheme-phoneme correspondences has little impact on reading fluency when phonological skills are poor. Children with good phonological skills at pre-test benefited more from the exposure to GraphoGame-NL than children with poor phonological skills. We suggest reducing the weekly playing intensity once letter-sound knowledge accuracy reaches ceiling, and extending the overall training period. This might allow poor performers to get more out of the game, especially to give more time for maturation of phonological skills (see *Borleffs et al., 2018* for a similar suggestion based on data from GraphoGame for Standard Indonesian). Future studies should furthermore focus on identifying factors that best contribute to training phonological skills. In Appendix 1, we provide a detailed description of the games used in this research, as we believe this to be crucial in enabling future research to uncover the mechanisms of (more) successful interventions.

## Question 4: assessment tools

Our final research question focused on the impact of assessment tools on examining intervention effects, *i.e.*, in-game assessment tools *vs*. pen-and-paper assessment tools. With in-game assessments, we were able to detect intervention-related improvements in letter-sound knowledge; however, these assessments could not confirm the results we found for reading fluency using pen-and-paper tests. We note that the implementation of our in-game written lexical decision task had poor reliability and only weak associations with other variables (see Methods of Written lexical decision). Therefore, it might be an unsuitable tool to capture reading skills, at least in this group of beginning readers. Another possible reason why an in-game effect for reading fluency was not observed can be seen as a question of the distance of learning transfer. Measuring (timed) letter-sound knowledge before and after a training of letter-sound correspondences can be considered a near training transfer because both are closely related, whereas evaluating changes in reading fluency based on a combined training of letter-sound correspondences and phonological awareness can be considered a far learning transfer. These skills are not directly related and intermediate developmental steps (*e.g.*, syllable recognition) might be required, which have their own cognitive prerequisites and developmental timelines (*Froyen et al., 2009*; *Vaessen & Blomert, 2010*). Indeed, near training transfer is reported almost unanimously for letter knowledge (*Richardson & Lyytinen, 2014*) as this skill is easily trainable and measurable, while far learning transfer improvements in phonological awareness and reading fluency are the exception, as they are expected to take longer to

train and yield overall smaller effects (*e.g.*, studies reported in *McTigue et al., 2020*; *Carvalhais et al., 2020*; *Lovio et al., 2012*; *Ktisti, 2015*).

In addition to learning transfer, how reading and reading-related skills are measured in intervention studies can have a considerable impact on the conclusions. This is also important in our findings. When analysing the letter-sound knowledge task at the level of the individual letter, we find intervention-related gains which were not captured using aggregated data from the same task, which would arguably correspond to a conventional pen-and-paper assessment. In the case of letter-sound knowledge for example, pen-and-paper tests are typically administered without time pressure and reach ceiling within the first few months of school (*Blomert & Willems, 2010*), thus losing predictive and evaluative power. By administering a speeded letter-sound knowledge task which measures response times on top of accuracy, the fluency with which letter-sound associations are retrieved from memory can also be assessed. Such a task is indeed more specifically related to the fluency of multimodal processing of audio-visual information (*Blomert, 2011*; *Hahn, Foxe & Molholm, 2014*). Thus, these in-game assessments, by measuring accuracy and response times at the item level, tap into the domain of automatisation to an extent which conventional pen-and-paper tests are unable to capture. Our study therefore demonstrates that the evaluation of response to intervention depends on the choice of outcome measure and the accompanying assessment tool and the statistical analysis. We further showed that in-game behaviour in serious games provides potentially sensitive measures to dynamically assess (pre)literacy skills.

## Limitations

We acknowledge several limitations in the design and procedure, which should be considered when interpreting our results and analyses. The unexpectedly large pre-test differences forced us to split our sample by country, which led to smaller groups and reduced power compared to the study we initially conceived. Due to significant group differences at T1, we cannot rule out regression to the mean as a possible explanation for some of our effects. The analyses also tested the inclusion of a wide range of measures as covariates in a conservative, yet exploratory fashion. We highly recommend replication of our study with other cohorts of Dutch and Flemish children.

An additional weakness is that we only measured reading fluency at T2. Due to an earlier pilot showing floor results and due to time constraints for testing at schools, we decided not to collect such data at T1. As a result, we could not directly test interactions between reading fluency improvement and other factors. However, by controlling reading fluency outcome for reading precursors at T1 (letter knowledge, phonological awareness, rapid automatised naming, and age), these results are nevertheless relevant and meaningful.

Another issue arises from the fact that the teachers who participated were favourable, or at least open, to the use of digital tools in their classrooms, and were furthermore not blinded to the gaming conditions. Thus, they knew their treatment allocation. This may have changed their teaching style, which is difficult to control or correct for. To balance the impact single classrooms may have on intervention effects, children should ideally be

randomised individually (for example, one third of a classroom playing the reading game, one third playing a control game and one third not playing). From our experience, this is hard to implement in classrooms and it would also negatively affect classroom atmosphere if some children were not allowed to play. An alternative is to implement the playing at home, which would come with its own set of challenges like how to ensure daily playing or prevent excessively long gaming sessions (*Ronimus & Lyytinen, 2015*).

Finally, the math game may not have been the best control condition. Through data collected with an auditory EEG paradigm from a subset of the children in the present study, it became apparent that playing the math game might also contribute to the development of phonological awareness skills (*Glatz, 2018*). As arithmetic representations are also phonological in nature (*De Smedt & Boets, 2010*; *De Smedt et al., 2010*), both games ultimately promote careful listening and fast access to phonological representations. Future research on computerised literacy training should therefore implement an active control condition where the improvements of video gaming can be expected in the visual or motor domain rather than in verbal and/or auditory learning.

## CONCLUSION

We conducted one of the first literacy digital game-based learning studies relying on single-trial data from in-game tasks to evaluate its effectiveness. Playing GraphoGame-NL led to an increase in mastery of grapheme-phoneme correspondences and to small improvements in reading fluency. Demographic characteristics such as familial risk of dyslexia or languages/dialects spoken at home had little impact on response to intervention. This study presented evidence that GraphoGame-NL can serve a diagnostic function and thus replace or extend assessment by means of conventional tests of literacy skills.

### Funding

This research was supported by the Netherlands Organisation for Scientific Research (NWO) and the Graduate School of Behavioural and Cognitive Neuroscience (BCN) (No. 022.004.008). We received financial support from the Open Access Publication Fund of Charité–Universitätsmedizin Berlin and the German Research Foundation (DFG) for the APC. The funders had no role in study design, data collection and analysis, decision to publish, or preparation of the manuscript.

### Grant Disclosures

The following grant information was disclosed by the authors:
Netherlands Organisation for Scientific Research (NWO).
Graduate School of Behavioural and Cognitive Neuroscience (BCN): No. 022.004.008.
Charité–Universitätsmedizin Berlin.
German Research Foundation (DFG) for the APC.

## Competing Interests

The authors declare that they have no competing interests.

## Author Contributions

- Toivo Glatz conceived and designed the experiments, performed the experiments, analyzed the data, prepared figures and/or tables, authored or reviewed drafts of the article, and approved the final draft.
- Wim Tops conceived and designed the experiments, authored or reviewed drafts of the article, and approved the final draft.
- Elisabeth Borleffs conceived and designed the experiments, authored or reviewed drafts of the article, and approved the final draft.
- Ulla Richardson conceived and designed the experiments, authored or reviewed drafts of the article, and approved the final draft.
- Natasha Maurits conceived and designed the experiments, authored or reviewed drafts of the article, and approved the final draft.
- Annemie Desoete conceived and designed the experiments, authored or reviewed drafts of the article, and approved the final draft.
- Ben Maassen conceived and designed the experiments, authored or reviewed drafts of the article, and approved the final draft.

## Human Ethics

The following information was supplied relating to ethical approvals (*i.e.*, approving body and any reference numbers):

The research ethics committee (CETO) of the Faculty of Arts at the University of Groningen, as well as the Ethical Committee of the Faculty of Psychology at the University of Ghent (Application Ref: 2015/25) approved the research protocol.

## Data Availability

The raw data and scripts are available in the Supplemental Files and at OSF: Glatz, Toivo. 2023. "Open Data of a Cluster-Randomized Trial of GraphoGame-NL in Groningen, The Netherlands and Ghent, Belgium." OSF. June 5. DOI 10.17605/OSF.IO/4P8HZ

## Supplemental Information

Supplemental information for this article can be found online at http://dx.doi.org/10.7717/peerj.15499#supplemental-information.

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
