# Peer review of "Dynamic assessment of the effectiveness of digital game-based literacy training in beginning readers: a cluster randomised controlled trial"

_PeerJ, doi:10.7717/peerj.15499_

## Round 0.1 · original submission · Major Revisions

Dear Toivo and team,

Thank you for submitting your study to PeerJ for peer review. I understand how difficult it is to carry out reading intervention trials in schools, and hence can appreciate how complex and challenging this study may have been to execute. I also understand how some of that complexity can lead to methodological limitations for practical reasons. Such challenges mean that reading intervention trials are rarer than reading researchers would like - which is why I would like to suggest that we continue to work together to see if we can get the manuscript into a publishable state.

Prior to receiving the reviewers' comments, I reviewed the manuscript myself. I agree with all the reviewers that this study is potentially important. I also agree with reviewers 2 and 3 that much work needs to be done before we can fully determine if it is publishable. The reviewers have gone to considerable effort in providing suggestions or asking questions to help you modify the manuscript to address its current limitations or ambiguities. I believe that following their suggestions will address most of the questions/concerns that I have from my own review. However, I do have a few additional suggestions that may help you.

First, in line with Reviewer 2 and Reviewer 3, I am concerned that the controls were all recruited from Belgium schools. As you note in your manuscript, Belgian children performed significantly below Dutch children on all pre-test measures. It is therefore highly likely that they would perform below Dutch children on the post-test measures - regardless of any intervention effects. Thus, this is an inadequate control group.

I would strongly suggest that you get together with your team and discuss the issue of the inadequate control group before you do anything else - since the study is unpublishable without addressing this issue. You do have some options. First, thankfully, you have your maths control group, which if I understand correctly, comprises both Belgian and Dutch children (Note: I agree with reviewers 2 and 3 that you need to clarify your randomisation procedures). So, you could simply delete the inadequate control group and use the GG and Maths groups. Alternatively, you might want to divide this study into two studies: one comparing GG and Maths groups in Dutch, and one comparing GG, Maths, and Control in Belgium (if you had equal and adequate groups). There may be other options as well. So, again, I strongly suggest that you resolve this issue with your team before you do anything else on this manuscript. If you cannot resolve it, it seems unlikely that we could publish it.

Second, I again agree with reviewers 2 and 3 that your methods and results are unclear. Given this is an intervention study, I strongly suggest you follow the CONSORT guidelines for reporting the methods and results. This should include a flow diagram that shows the "flow" of participants between the different phases of the study. If you follow the CONSORT guidelines, you should address a number of requests for clarification by the reviewers.

Third, I believe the clarity of the manuscript overall could be approved in two ways. I think your questions need to be tightened up. In fact, I would suggest that you rephrase your questions as succinct aims, providing hypotheses, where possible, from previous research. Once you have your aims clear, I suggest you restructure your Results and Discussion sections around those aims. That is, include subsections in both of these sections that address each of your aims explicitly.

Fourth, a minor issue that I had with the manuscript was the use of the term GraphoGame. Is this some kind of brand, or software, or something else? I am confused about what GG actually is because I get the impression from the Introduction that GG has been adapted or used in so many different ways that it could equally be called something more general like "early reading intervention". In a nutshell (and in modern speak), is GG "a thing"? And if it is, what is it? What makes an intervention a GG intervention and not some other type of intervention?

As I mentioned above, in addition to these broader issues/suggestions, I recommend that you address as many of the reviewers' suggestions as you can, since this will significantly increase our chance of getting this study into a publishable form.

With best wishes (and good luck),

Genevieve

·

Basic reporting

The writing is clear, unambiguous and professional. The article does include sufficient introduction and background to demonstrate how the work fits into the broader field of knowledge. This being said, I was a bit surprised that the seminal review article by Ziegler & Goswami (2005) was not cited in lines 17-20. Also, I was surprised that the studies by Landerl et al. (2013) and Ziegler et al. (2010) were not cited in lines 36 to 39 in favor of the claim that the weight of the classic language predictors (PA, RAN, vocabulary) varies systematically with the complexity of the orthography. Line 60-61: Please note that there is a recent French version of GG that has been used and tested for early detection and remediation of reading difficulties in a school setting (support groups). Unfortunately, the publication is in French but it is in a real journal [Ruiz, J.-P., Lassault, J., Sprenger-Charolles, L., Richardson, U., Lyytinen, H., & Ziegler, J. C. (2017). GraphoGame : un outil numérique pour enfants en difficultés d’apprentissage de la lecture [GraphoGame: a digital tool for children with difficulties in learning-to-read]. Approche Neuropsychologique des Apprentissages chez l’Enfant (A.N.A.E.), 148, 333-343]. Ruiz et al. also used a math game against with the GG progress was assessed (lines 68-69). Finally, Ruiz et al. also used a within-game progress assessment. At the beginning and at the end of each stream, they used an identical “summary-test-level”, which allowed them to assess whether a given child made significant progress during the stream. Indeed, they showed that the within-game progress was highly significant after playing GG (see Figure 2 of the attached article). So, technically speaking, the claim in lines 485-486 “To our knowledge, we are the first to use and report on single-trial in-game” is not quite correct (although I don't care much about primacy). One final point: I didn’t see any links as to how the raw data has been made available in accordance with PeerJ Data Sharing policy. The figures are very nice but the figure legends need to be improved. In figure 4, it is not clear what “-1z/+1z items seen” refers to.

References cited
Landerl, K., Ramus, F., Moll, K., Lyytinen, H., Leppanen, P. H., Lohvansuu, K., . . . Schulte-Korne, G. (2013). Predictors of developmental dyslexia in European orthographies with varying complexity. J Child Psychol Psychiatry, 54(6), 686-694.

Ziegler, J. C., Bertrand, D., Tóth, D., Csépe, V., Reis, A., Faísca, L., . . . Blomert, L. (2010). Orthographic depth and its impact on universal predictors of reading: A cross-language investigation. Psychological Science, 21(4), 551–559.

Ziegler, J. C., & Goswami, U. (2005). Reading acquisition, developmental dyslexia, and skilled reading across languages: a psycholinguistic grain size theory. Psychological Bulletin, 131(1), 3-29.

Experimental design

The investigation was performed rigorously and to a high technical standard. The design was very good especially the fact that two control groups were used (math and passive). Also, the sample size is rather impressive. The methods are described with sufficient detail and information to replicate.

There is one shortcoming that should be acknowledged, which is the fact that schools were not selected randomly, which leaves the possibility open that only those schools participated in the study, which had a positive attitudes towards using digital tools in education. Similarly, classes should have been assigned to treatment conditions in a random fashion to avoid potential biases (e.g., the reading treatment was given to those teachers who were the most favorable towards using digital tools).

Validity of the findings

Data are robust and the statistical analyses are very professional and state-of-the-art (e.g., mixed effect models etc.). I didn’t see how the data on which the conclusions are based are made available in an acceptable discipline-specific repository. The conclusion are well stated, linked to original research question & limited to supporting results. The conclusions are appropriately stated and connected to the original question.

Additional comments

Overall, this is a very nice paper !

·

Basic reporting

A training study is reported in which a 5 to 7 week intervention at the beginning of 1st grade are playing an adaptive training software for reading (Graphogame). Training effects are compared with a treated control group playing a similar game with mathematical contents an an untreated control group. In the post-test, children in the GG group read faster than the passive group, but did not differ from the maths group. They also new more letters than the passive group, while no effects on phonological awareness could be demonstrated. Inspection of participant and training characteristics indicated that girls who had high PA at pretest played most intensively and accordingly showed the largest improvement in reading fluency.

Reporting the results of intervention studies is highly important, even if (or especially if) the intervention can only partly produce the expected effects. The study is based on an impressive sample of 250 children from 16 classrooms. The inclusion of two control groups is worthwhile, and so is the investigation of the effects of how (intensly) the software was used. This information is highly valuable in order to further develop the software itself and specify as to how it can be used in schools (i.e. is it helpful for struggling readers to catch up with peers or is it more useful in supporting teachers to get childrenwho are interested in reading to practice their letters and reading).
However, I have a number of questions that should be dealt with before the paper is considered for publication:

Introduction:
L 15 - 20: Is it the prevalence or the nature of reading problems that differs between languanges? I am highly sceptical that it is both as is suggested here. When the very same diagnostic criteria are applied (e.g., low reading accuracy), prevalence will differ. However, countries use different tests and different criteria in order to identify their struggling readers. If we select the lowest 10 % in a standardized reading test (with or without discrepancy to IQ – this is an ongoing discussion), prevalence rates would necessarily be the same, but manifestations might differ.
L. 19: Seymour et al. (2003) does not report data on the nature of reading problems.
l.78: what exactly was the „personal communication“ part in the description of Blomert and Willems (2010) – the study is published, so which information was received from authors that is not in the paper?
l.82: „FR“ is not introduced

Method and Results
- While the manuscript describes the material used in the training software (l.199 ff), I could not find any description of the activities included in the reading and the maths version, the number of levels that can be played or how the number of distractors adapts to the performance level of the child, which makes it quite difficult to understand the section on in-game exposure measures.


- In general, I found the structure of the study report hard to follow. The Method section describes the measures in a completely different order from the Results section. The Data analysis section preceding the Results section describes the statistical analysis oft he pre-posttest measures, while the description of analysis of in-game measures appears only later in the Results section. While letter-sound integration is presented as an in-game measure in the Methods section, it appears before the analysis oft he in-game measures in the Results section. Same fort he written-lexical decision task, which is presented as the last task in Methods but appears right after Reading Fluency in Results, followed by the offline measures of PA and then the LSSI tasks, which are agin in-game measures. It took me quite a while to work out what was really done and I am still not sure that I understand everything.

Discussion:
- The discussion also should be re-structured: Earlier literature was presented in the Introduction, but the very same studies are presented in the discussion again, with more detail. In my understanding, the literature review is part of the Introduction, while the discussion compares current results with earlier literature. Is it correct that there is not a single study showing that GG can improve phonological awareness? This would be important to know, particularly in the context of the current study which seems to suggestthat children who have good PA to start with profit more than children with poor PA. It would also be helpful if the literature review in the intro would be structured based on constructs (for instance: LK, PA , reading – what ist he evidence on GG training effects? The

- To my knowledge, the original concept of GG was to help children with delayed development in letter knowledge to overcome these early hurdles (as LK was the most important predictor in the Jyväskylä Longitudinal Study). To what extent would authors say that this was achieved? In my reading, GG seems to be an attractive gaming software for those who are interested in letters and reading and already have some basic knowledge. What would be needed to support those who struggle with these early steps? Towards the end oft he manuscripg (l.597ff) authors suggest that they need more time in order to play the full game. However, this is not tested in the current study, so such conclusions must be drawn with great caution.

Experimental design

- Please explain why it was chosen to assign full classes to training conditions and to what extent this assignment was randomized. Ideally, one third of students in each classroom should have been assigned to each training condition randomly, which would have controlled for the aparent differences between classes and countries. Having no Dutch students and classes in the passive condition seems to cause immense problems for the interpretation of training effects (see next point).

- I also think that it is unfortunate that reading was not assessed at pre-test, as it makes it impossilbe to interpret the group differences in reading fluency at post-test. As reading preparation seems to be more intense in NL than in Belgium, it is highly plausible that the passive group, which consisted of Belgian children only, included a higher number of non-readers at the start oft he training, who may have gained just as much in their reading skills during the study period as the two other groups. There are more and more studies that use 1-min test formats at the onset of schooling. Even total nonreaders are willing to inspect letter patterns for this short period and the readers are easily identified. This is becoming standard in 1st grade studies (e.g., Caravolas et al.2012, 2013; Peterson et al., 2017).

- While the manuscript describes the material used in the training software (l.199 ff), I could not find any description of the activities included in the reading and the maths version, the number of levels that can be played or how the number of distractors adapts to the performance level of the child, which makes it quite difficult to understand the section on in-game exposure measures.

- How exactly was family risk defined? Who are „close relatives“ (l.185/186) and were they required to have a dyslexia diagnosis or was it sufficient to reprort unspecific reading problems in the questionnaire? The fact that there were no major differences may be related to small sample size (as is argued in the paper) as well as to lack of sensitivity of the selection criterion?

l. 193: What is „below average intelligence“?
l. 235: Why was it considered important to include RAN? If it is correct that this measure was assessed in pre- as well as post-test, then please report a pre-posttest analysis. I noticed that RAN is not even mentioned in the discussion.
l.252: Please provide more info on the words in the reading fluency conditions: length, frequency, bigram-frequency…
l. 268: Please provide more info on the 16 words and 16 pseudowords (e.g., length, bigram-frequency) and on how the task was scored.

Validity of the findings

Overall, statstical analysis is sound and based on up-to-date approaches. Particularly the discussion will need some clarification as I am not always sure, which part of the analysis the different summaries of results are related to.

l.323ff: Please refer to Table 2 when group differences are discussed. Handedness should be reported in Tabl. 2
l. 406-7: How was „progess“ and „motivation towards the game“ defined in this analysis? RES results should be reported.
l. 427: What are „post-test reading scores“? Is this only the reading fluency measure or was lexical decision included?
l. 460f: „…led to higher reading skills in the girls who had above-average PA skills at pre-test“ – but only if they played extensivley, right? In the first analyses, gender did not enter the model.
l. 463: What is a semi-transparent orthography?
l.523 – 529: Please restructure sentence.
l.546f: „Girls with good pre-test PA skills attained higher reading fluency scores after playing the reading game than their opeers in the other two groups…“ Perhaps I lost track, but the gender effect was only reported for the exposure measure and I thought that the analysis on exposure was limited to the reading group?
l.555: Same – gender did not appear as a predictor for the pre-posttest assessment of reading fluency.
l. 588: Moll et al reports a cross-sectional analysis and is therefore not informative with respect to „later“ reading proficiency
l. 607 „those with low pre-test LK and PA skills“ – which analysis tells me that children with low LK and low PA profited more from GG?
l. 607: „vast majority of children indicated to have enjoyed playing the game a lot“ – this is not reported in Results.

·

Basic reporting

no comment.

Experimental design

The research questions are well defined. There are however some limitations in the design as presented in the manuscript:
- Selection (inclusion) procedure and randomization procedure are not clear, please explain in more detail.
- Please explain why you choose the specific study design to test your three research questions.
- Please provide a more detailed description of the tests used.

Please see section 4 (general comments for the authors), subsection "methods", for more detailed comments and suggestions on the design)

Validity of the findings

The study presents a well-designed plan for analyses. However the presentation of the results is difficult to read, and would benefit from a thorough revision.
Please find a detailed list of comments and suggestions on the reported findings in section 4 (general comments for the authors), subsections "results", and "discussion".
Some general remarks:

- The sample size as used in the analyses is unclear to me. The authors note a sample size of 250 with incidentally 2 missing values. But in the datafile, e.g. for LK-T2 missing data seems present for about 70 subjects. Please explain.
- It would be insightful to have a Table with an overview of correlations between the outcome variables (PA, reading, lexical decision, etc)
- It is often unclear to me which specific scores are entered in the analyses. E.g. PA refers to? All separate PA measures, a composite measure? Please make clear what scores are used either in the results or in the method section.
- How did you control for multiple comparisons?
- Please explain the (a priori) reasons for inclusion of variables in the models. E.g. besides condition, for the Reading Fluency T2 model you entered LK, PA, RAN colors, handedness and country, whereas for the Lexical Decision T2 model, Age, LK, and country were entered. I have difficulties understanding the rationale/consistency behind this.
- If possible, it would be appreciated if the authors could provide measures of effect size for the reported significant effects to help interpretation.

- It would be helpful if you organized your discussion along the line of your research questions (Question x = …, this was examined by …, Results revealed …).
- Please be careful with statements on reading progress, as you only measured reading at posttest (T2).
- As many findings seem to be of a explorative nature, I suggest to recommend replication of findings as an important step to be taken.

Additional comments

This study aims to evaluate the effectiveness of Graphogame (GG), a game-based reading training. In particular the objective is to understand conflicting results between GG-studies in the past, and to provide a comprehensive window on the factors that potentially influence GG outcomes. This results in three research questions:
(1) Are in-game proficiency tests more sensitive to training-induced changes than conventional offline tests?
(2) Which subgroups of children benefit most from GG?
(3) Which training characteristics (e.g. content, duration) contribute to the effectiveness as measures by reading fluency and lexical decision

I found the goals and background of the study are relevant and interesting. Especially the focus on in-game indices that are sensitive to more micro-level intervention-induced changes is an interesting approach to provide a more detailed window on the nature of learning processes during game-based reading intervention, than is usually presented in the pertinent literature.

Please find below my comments and suggestions in detail:

Abstract
- Please include information on sample size in the abstract

Introduction
In general, I advise to polish the introduction in order to be more precise in the statements made. In the list below, I presented several examples of aspects that were unclear to me.

- L.8-9. Definition of dyslexia. I recommend to include a more precise definition that refers to specific and persistent (e.g. DSM-V) problems, and please include a reference.
- L. 15. “.. with the prevalence of dyslexia necessarily differing across .. ” Why necessarily? Please make your statement more explicit, e.g. prevalence rates appear to differ as a function of orthographic depth and .. etc
- L. 44. Please include references supporting the promise of edugames in reading instruction.
- L. 45. I recommend to rephrase here, in the line of: ‘Several characteristics seem to contribute to the potential of serious games: etc.’ Please reformulate the list, it is a bit vague now. I recommend to also include some references of literature from the field of serious game design itself (e.g. Kiili, De Freitas, or Prensky).
- L. 63. “literacy crisis”. Which crisis? Unfortunately, the word crisis is used regularly in the field of literacy. So please provide a short explanation + reference.
L. 66-71. Who achieved more gain? Poor readers, or general population of (early) readers, or prereaders…
L. 73-76. Were these children with speech disorders, or with reading disorders or both?
L. 87-90. Although I am sympathetic with the assumption that visual specialization in these areas is driven by letter-speech sound learning, this is open to debate and it seems safer/more neutral to replace “during letter-sound-correspondence learning” by "visual word recognition".
L. 94. .. did not reach significance due to … Obviously, you do not know that for sure, better to say …. possibly due to …
L. 97. You note that studies with positive findings differ from those without on several methodological aspects. This is very interesting. Please explain.
L. 113-115. This might be more than only ceiling levels. In the pertinent literature, letter knowledge (explicit knowledge about the name or sound of a letter) is often considered as a (partly) different process than speeded or fluent letter-speech sound integration (more implicit multimodal processing of audiovisual information). See e.g. Blomert, 2011; Hahn, Foxe, & Molholm, 2014.
L. 146-147. I advise to delete this sentence, instead of this bold statement, it is far more interesting to know which specific factors that might explain differences in the outcomes of previous results, will be examined. These factors are described in L.150 ->, making this sentence redundant.
L. 148, 153. “Gauging” seems a somewhat unusual term to me, you might consider replacing it by indexing, tapping, measuring

Method
General
- Selection (inclusion) procedure and randomization procedure are not clear, please explain in more detail.
- Please explain why you choose this specific design to test your three research questions?

In detail
L. 170. How was “willing to motivate and enable each student” operationalized?
L. 172-173. Again, how was this operationalized? How was it judged a priori whether the teacher was able to do so?
L. 173. Important for generalizability of findings, do you have information on how many schools were excluded by the selection criteria? Especially b, and d.

L. 175. It is important to know how subjects were allocated to condition, especially because for several measures no pretest was administered. Therefore, it is strongly recommended to explain the randomization procedure in detail, including what is meant by 'semi'. Note, it is also advised to note that the teachers were not blinded to condition. Were the researchers blinded?
L. 205. ‘over’ = ‘to’
L. 229. Please include reliability information for that tasks used.
L. 237, 242. How many tests did you use exactly for PA1 and PA2? Why did you choose to use so many tests for PA? What is the score? Accuracy?
L. 261-270. Letter-speech sound tests, and lexical decision task. Please provide a more detailed description of the tasks. As it is now, the nature of the task, as well as the score or scores used for analysis remain unclear. Note: I recommend to replace “written lexical decision” by “visual lexical decision”.
L 271. Procedure. Please explain why you decided to test reading and lexical decision only at T2.


Results
General:
- Please provide info on whether data met the test assumptions.
- The sample size as used in the analyses is unclear to me. The authors note a sample size of 250 with incidentally 2 missing values. But in the datafile, e.g. for LK-T2 missing data seems present for about 70 subjects. Please explain.
- It would be insightful to have a Table with an overview of correlations between the outcome variables (PA, reading, lexical decision, etc)
- It is often unclear which specific scores are entered in the analyses. E.g. PA refers to? All separate PA measures, a composite measure? Please make clear what scores are used either here or in the method section.
- How did you control for multiple comparisons?
- Please explain the (a priori) reasons for inclusion of variables in the models. E.g. besides condition, for Reading Fluency T2 you entered LK, PA, RAN colors, handedness and country, whereas for Lexical Decision T2, Age, LK, and country were entered. Maybe I am missing the point, but I have difficulties understanding the rationale/consistency behind this.
- If possible, it would be appreciated if the authors could provide measures of effect size for the reported significant effects to help interpretation.

Details:
- L.323-324. Please refer to Table 2. Q: Did you test whether or not the groups were balanced? If so, please inform us on the test statistics.
- L.335. What do you precisely mean by PA? As you used several measures to tap PA. Same for LK.
L. 358-360. Please explain the effect in more detail. It is unclear to me whether this trend refers to an interaction-effect between condition and time, favoring the reading group, or simply a trend for an effect of time for the reading group.
L.362. I assume that LK and LSSI refer to the same tests, right? If so, please use one term to avoid confusion.
L. 366. ‘absolute LK’: what is this? And how does it differ from ‘LK’
L. 377. How does these 8809 RT trials relate to the 14,613 accuracy trials?
Are all other trials incorrect responses? Or did you also remove oultiers: if so, please inform us about percentage removed. It is also informative to inform about percentage correct at T1 vs T2.
L. 380-381. What is the difference between ‘trial number’ and ‘ stimulus presentation order’?
L. 386-287. Interesting! Did T1-T2 growth in acc predict growth in RT?
L. 404-408. How did you test this? Please explain and provide details on statistics.

Discussion
General
- It would be helpful if you organized your discussion along the line of your research questions (Question x = …, this was examined by …, Results revealed …).
- Please be careful with statements on reading progress, as you only measured reading at T2.
- As many findings seem to be of a explorative nature, I strongly suggest to recommend replication of findings as an important step to be taken.

Details
L. 471-474. “In addition to their potential limitations with regard to test-retest reliability”, it is good to note here that reliabilities in the range .71-.86 are usually considered acceptable in group comparisons, and that this statement is somewhat problematic as we have no information on reliability / validity of the measures used in this study.
L. 476. “lack of sensitivity of change”, I believe this discussion is more a question of measuring far vs near transfer effects of training than one of sensitivity per se.
L. 487-488. We would very much like to see a more in depth discussion on the findings on game-induced changes in letter-speech-sound-associations, as this is the primary focus of the graphogame. Please differentiate between accuracy and speed. To me, whereas the accuracy effect seems graphogame specific, the effect on RT seems more an effect of experience with gameplay as both gaming conditions improved as opposed to the no-game controls.
L. 501-503. I fully agree, this is an interesting approach to come to more fine-grained insight into intervention dynamics.
L. 505-507. This phrase is somewhat confusing, as it suggests you included a specific sample of children performing an arbitrary cut-off criterion.
L. 541-544. I do not understand the function of this statement here, especially since you just mentioned your sample size of FR children was very small (n = 7-18). Please delete or rewrite.
L. 555. “..it promoted reading skills only in girls with good pre-test PA scores.” I believe your design doesn’t allow you to say this, as you did not measure reading at t1. Moreover, I assume that those girls with better PA at t1, were probably the ones with better reading at t1, i.e. PA is an index of t1 reading skill. Note: isn't it possibly the lack of variance in PA in boys that explains the lack of correlation with reading at t2 for them?
L 558-561. “…we feel our sample was representative … “ I believe you cannot substantiate this feeling. To me it is far more plausible that it is not representative as the sample of schools is limited and geographically centered around only two locations. At least for NL, it is fair to say Groningen is not representative for the Dutch population. I recommend to remove this sentence from the manuscript.
L 591-599. See comment on L.555.
L. 600. Presumably. I suggest to replace by “Possibly”.
L. 605-607. See comment on L.555.

---

## Round 0.2 · Major Revisions

I would like to start by commending authors for the amount of work that has obviously been put into revising this manuscript in response to suggestions by the three reviewers and the editor. It is quite an achievement.

Two of the three original reviewers have read the revised version (Note: the third reviewer has deferred to the editor to make a decision). They are generally pleased with the changes, which have left them with only minor suggestions for further improvement. I recommend that the authors make as many of these changes as possible, and provide clear explanations/justifications for those that are not made.

The editor - that is, I - acted as both editor and reviewer for the original version of the manuscript. I think it is important that I continue to do so since a significant part of the revision stemmed from my own comments or suggestions - particularly in relation to the analysis. The revised version of the manuscript leaves me with more comments/suggestions than the two reviewers. Most of these suggestions are minor, but a couple verge on being major. Since I cannot classify these as "verging on major" in the PeerJ system, I have classified them as major overall, but I believe they are achievable. In my comments below, I start by outlining the more major suggestions, and then the minor ones.

I wish the authors the best of luck with the revisions, and I look forward to seeing the next version in due course.

More major
1. Because this manuscript reports on an intervention study, it is going to be of interest to people beyond academia. It is therefore important that it is simple to understand. I struggled to understand a number of sections in the manuscript, which raises the issue of how understandable it will be to others. I think the readability of the manuscript could be improved markedly with some uniform re-organisation of information within each subsection of the manuscript that makes the order of information easier to predict by the reader, and hence to follow. To achieve this, I suggest the following:

i. Combine the aims and hypotheses in the final section of the Introduction to make the aims clearer (Note: remove any reference to questions). For example, "The first aim of this study was to identify the characteristics of children who benefit most from GG-NL ("child characteristics"). From previous studies, we hypothesised that X, Y, and Z". Then do the same kind of thing for the second and third hypotheses.

ii. Given that you only have three aims, focus solely on these three aims in the Introduction (ie remove reference to the interactions). This will leave you with three foci: (1) child characteristics, (2) assessment types, (3) training properties.

iii. In the Introduction, in the paragraph that relates to the first aim (child characteristics; starting line 119), previous studies are criticised for the use of cutoffs or teacher reports. This is not a strong argument since all recruitment procedures in general have their limitations (including the current study), and also because some recruitment procedures are perfectly OK for certain questions (e.g., indentifying children using teacher ratings is perfectly OK if your question is: "how well do teacher-identified children respond to GG?"). So, rather than criticising these studies, I would suggest using the methods and results of these studies to formulate/justify an hypothesis for the first aim.

iv. Do the same for the third aim (training properties). At the moment, it is briefly mentioned that previous studies have varied in these properties, and that these properties might be important, but no effort is used to look at previous research to make a specific hypothesis. Refer to previous research to identify which properties might be important, and make a relevant hypothesis. Previous studies do not necessarily have to be restricted to GG.

v. Having done iii and iv, you have set up a three-part structure (child characteristics, assessment types, training properties) that can be used to help simplify the presentation of information in subsequent sections. Specifically, under Assessment, create three subsections, and list the relevant variables measured. At the moment, it is not clear which child characteristics you measured and hence included in the analyses. You have the Assessments in there, so that presumable includes all the assessment types you are interested in. But I can't see how you measured training properties. So add a subsection for that, and outline how you measure the variables relevant to your aim and hypothesis outlined in the Introduction.

vi. Do the same for the Data analysis section. That is, subdivide this into three sections (child characteristics, assessment types, training properties) and explain what analyses you used to address each aim, what those analyses tell us, and what we would expect if the hypothesis for each aim were correct.
vii. Do the same subdivision for the Results section.

vii. And do the same subdivision for the Discussion.

Hopefully, this uniform structure across the manuscript will make the flow of information more predictable for the reader, and hence easier to follow and understand.

2. I struggled to understand the analysis section. I think this is partly because it was not clear which analysis was being used to address which aim of the study (hopefully the above suggestions will help clarify that). It is also because I am not familiar with the types of analyses that you used, which are not typical for intervention studies. There is nothing wrong with using a different analytical approach, but this does mean that it is even more important to explain why a certain analysis was used, what it was designed to tell us, and what to expect if the hypothesis is supported. I suggest adding this information to the Data Analysis section in each of the three suggested subsections in turn (child characteristics, assessment types, training properties).

3. I also struggled with understanding the Results section, for the same reasons as the Analysis section. Hopefully, this will be resolved by your response to 2 above. However, please provide any extra information you can think of that will allow you to gently guide the reader along the way so they don't get lost in technical statistical jargon. When I struggle with this, I revert to back to a step-by-step process (e.g., In the first step of the analysis we did X to examine Y. In the second step, we did Z to test A, and so on. This may or may not work for you).

4. In the Discussion, there are words used around the outcomes for reading fluency that are a bit misleading. For example, on line 659, it is said that "Some children with good pre-test PA skills attained higher reading-fluency scores after playing the reading game extensively". This suggests that the reading game improved the reading fluency scores. But we do not know this because the reading fluency scores of the children were not measured prior to training. So the wording around the interpretation of the reading fluency outcomes must be revised with care.

I should mention that the reading science community - particularly intervention experts - will not like such claims. As you know, I also have reservations about the inclusion of the reading fluency test, since it was not administered at Test 1, and so we cannot know that there was no difference between the groups at T1 on reading fluency. However, I am reserving judgement about the inclusion of reading fluency until the Analysis and Results sections are revised in a way that means I can understand them clearly. I can certainly see a reason for including T2 reading fluency for a predictive analysis - just not direct group comparisons at T2. We can tackle this issue at a later date once I better understand the new analysis.

Minor
Abstract: Just mention the final number of participants in the analysis in the Abstract.

Abstract: "on the classroom level" - please revise - meaning unclear.

Throughout manuscript: While the writing throughout the manuscript is generally clear, there are some sentences that are oddly phrased, and the use of commas is confusing in numerous places. It would be helpful if the manuscript could be carefully proofread by one or two experts in English and/or commas.

Throughout: letter knowledge (LK). I am confused about this. On line 28, letter knowledge is defined as "knowledge of grapheme phoneme correspondences". In fact, the latter is grapheme phoneme correspondence knowledge (often called GPC knowledge). In reading research, letter knowledge is the recognition letters (upper and lowercase) regardless of corresponding phonemes (/a/ /b/ /k/ /d/). There is also letter naming whereby children can name letters (e.g., "ay" "bee" "see" "dee"). So the skill referred to as letter knowledge in this study is not clear.

Throughout: There are a lot of acronyms used throughout the study. Write all terms in full. It makes it harder for a reader to follow the flow of an argument if they also have to remember the meaning of acronyms.
Throughout: The formatting of lists is not consistent throughout the manuscript. Sometimes i) is used, sometimes a) and sometimes bullet points. Have a look at PeerJ's formatting and use their formatting consistently throughout the manuscript.

Throughout: The formatting for numbers is also inconsistent (sometimes 4, sometimes four). Again, please refer to PeerJ articles to establish the norm and use consistently.

Assessment (line 236+): Do not use bullet points to demarcate different tests. See PeerJ for formatting for subdivisions.

Assessment: The tests are described at different levels of specificity (e.g. very little information describing abstract reasoning compared to reading fluency). For every assessment, please describe the stimuli, what the child was asked to do with the stimuli, how they responded, how their responses were scored, what those scores represented (e.g., total number of responses correct out of 10), and how to interpret those scores if appropriate (e.g., if less than X then fell below the average range).

Participants: Put this section at the start of the Methods.

Participants (last paragraph starting line 324): Remove this paragraph. You explain the situation elsewhere in proper detail. It is confusing present in isolation here.

·

Basic reporting

I have now read the revision and the authors took into account most of my comments and suggestions. I few minor points remain. Overall, I feel that the present paper makes an important contribution.

Lines 76-109. I am sorry to insist but I really think you should add a summary of the French GG intervention study (Ruiz et al., 2017). The sample size was rather good (N=34 in one study with 1 graders at risk of reading difficulties and N=35 for second graders), the study used an active control group in grade 1 and a passive control group in grade 1, it reports significant in-game progress measures on LSSI accuracy (measured before and after each stream), and the training effects generalized to reading outside the game, as measured by a 1-minute reading test in grade 1 and a standardized reading test in grade 2.

Experimental design

good

Validity of the findings

649-651. The outcomes revealed that these in-game single-trial response-time and accuracy tests yielded more sensitive proficiency measurements than conventional offline paper-and-pencil tests. What allows you to make this claim? How was the sensitivity compared? The two measures can’t be compared directly because the outside-the-game measures tend to measure generalization performance (e.g., fluency in a 1 min reading test), whereas the within-game measures (e.g., LSSI) are identical to what has been trained in the game.

Additional comments

I have now read the revision and the authors took into account most of my comments and suggestions. I few minor points remain. Overall, I feel that the present paper makes an important contribution.

Lines 92-96 : “Whereas medium-to-large effect sizes were found for reading, spelling, and PA for GraphoGame Rime compared to GraphoGame Phoneme, the difference did not reach significance, possibly due to the study's small sample size”. This sentence doesn’t make sense. If medium to large effct sizes were found of one intervention compared to the other, how can the difference between the two interventions not be significant. Do you mean to say that both interventions yielded medium-to-large effect sizes but the difference between them was not significant?

Lines 76-109. I am sorry to insist but I really think you should add a summary of the French GG intervention study (Ruiz et al., 2017). The sample size was rather good (N=34 in one study with 1 graders at risk of reading difficulties and N=35 for second graders), the study used an active control group in grade 1 and a passive control group in grade 1, it reports significant in-game progress measures on LSSI accuracy (measured before and after each stream), and the training effects generalized to reading outside the game, as measured by a 1-minute reading test in grade 1 and a standardized reading test in grade 2.

Lines 261-264. Not clear from the description how the LSSI task worked. The authors say that they measured “the amount of correctly named letters across the levels” but did the children really have to name letters? Also, it is misleading to say “test their letter knowledge” when they were presented with complex graphemes. Was this task really about letter knowledge or was this a “classic” GG level, in which phonemes were presented and children had to select the corresponding graphemes?
Figure 4 (legend). Please explain somewhere how you go from LSSI accuracy to log-odds scores? Please justify the transformation.

Line 617 “Unlike most previous GraphoGame studies (c.f. Richardson & Lyytinen, 2014), we but recruited”…delete “but”

649-651. The outcomes revealed that these in-game single-trial response-time and accuracy tests yielded more sensitive proficiency measurements than conventional offline paper-and-pencil tests. What allows you to make this claim? How was the sensitivity compared? The two measures can’t be compared directly because the outside-the-game measures tend to measure generalization performance (e.g., fluency in a 1 min reading test), whereas the within-game measures (e.g., LSSI) are identical to what has been trained in the game.

·

Basic reporting

The revised ms. is written in a clear and unambiguous manner. The authors resolved ambiguities in the introduction and added relevant references in the revised introduction. The authors provide a clear and relevant background, and argue how their study fits within a broader context of game-based training for reading acquisition. Raw data is shared in a well-organised manner, which is very helpfull. The research questions and hypotheses are explicitly stated.

INTRODUCTION - Comment:
- L. 105-107 (differences between previous studies).This type of interventions can be subject of relatively high attrition rates. Could this also be a difference between studies: do they differ in attrition rate, and, in particular, in the inclusion or exclusion of those dropping out of intervention into the analyses (thus, ITT analyses or not)?

Minor to very minor:
- L 19. “or” should be “and”.
- L. 20-21. Please include a reference for this statement. (e.g. Pennington)
- L. 201-204. What is your hypothesis concerning RQ3?

Experimental design

METHODS

The method secion is much improved, randomization and allocation procedure and many operationalisations are now much clearer to me. Nonetheless, I still have some recommendations to further improve this section:

- Thank you for including reliability estimates for the reading fluency task. Could you also include reliabilities for the other outcome measures?

- Are there pretest differences between the conditions (within country)? This seems relevant to know, considering no full randomization.

- I still found the descriptions of the LSS-identification and Written Lexical Decision, unclear. E.g., for LSS: is one letter(group) presented on the screen or also another as distractor? Do they also hear the speech sound or do they have to say the sound (or letter name?) themselves. Is item presentation response terminated or are visual stimuli only shortly presented? … E.G. for written lexical decision: Do they see all 16 items (or 32? As suggested in line 266) together on the screen? Or one by one? Do they really have to click on an icon on the screen with the mouse? Where are the response-icons presented, next to each item? How did you measure separate target reaction times (assuming each word is a target) when 16 targets are presented at the same time?
Please help us out here and provide a clear description.

- L. 382-386. Why did you include both raw scores and percentile scores of the same tests into the analyses to ad hoc decide which one to use based on their test statistics? Could you please give a rationale for this. Some people might otherwise think this method is promoting cherry-picking.

Validity of the findings

RESULTS

Again, the revised manuscript is much improved here. And again, some recommendations:

- For the LMM results, please include, next to the beta, the cofidence intervals (eg upper-lower 95%). It would also help to interpretate intervention results if you could include a Table with test-scores at T2.

- To avoid confusion on the part of the reader, please note which predictors were originally entered, or which (primary) predictors were dropped from the final model.

- PCA (l.565): please include the rationale for extracting one component only? (scree plot, eigenvalue criterion, other..).

Minor:
-L. 465. "RAN colours time". At t1 or t2?

DISCUSSION

L. 600-603. ".. and we observed a small to medium sized classroom wide benefit in terms of reading fluency compared to the other two conditions.". I am somewhat confused here, as you stated on L.427-430: "With regards to the effect of condition (see Figure 2), neither the reading (β = 0.27, t = 1.59, p = .115) nor the passive (β = 0.27, t = 1.63, p = .106) group differed from the math group, while the reading group read significantly more fluent that the passive group (β = 0.55, t = 3.18, p = .002)." Please reformulate / formulate more carefully (or explain).

Minor:
L. 737-748. You seem to hold the assumption here that PA unidirectionally impacts reading, while many studies show a reciprocal relationship between the two (or mostly a direction from early reading acquisition to PA development). Considering a more dynamic relation between the two might provide another view on this finding.

- Limitations. Due to practical contraints, randomisation was done by classroom, that is, children weren’t randomised to condition within classroom. In combination with the small sample size on the level of classrooms, this created the risk that idiosyncrasies of one or more classrooms might have obscured/interfered with intervention effects. The authors might want to point to this.

Additional comments

Many thanks for the thorough revision and the extensive reply to the points raised.

---

## Round 0.3 · Minor Revisions

Dear Dr Glatz,

Thank you for your resubmission of your article to PeerJ.
As you note in our response to the previous reviews, it has been quite a long time since this article was last submitted to PeerJ (over a year, I believe). In addition, you had obviously worked hard to make a lot of changes in response to the previous round of reviews. With both of these factors in mind, I did not feel I could make a decision on my own about the quality of the changes, so I contacted the two previous reviewers who offered up suggestions for improvements in the last round - in addition to my own.

Unfortunately, one of the reviewers was unable to obliged due to an unexpected personal emergency (such is 2020 and 2021). However, another reviewer was able to take this on. For the same reasons outlined above, this reviewer has read the manuscript "from scratch", as well as focusing in on responses to previous suggestions. The improved clarity of the manuscript has allowed the review to detect some more issues, which were obscured by lesser clarity previously, and hence there are a number of new issues/suggestions to address to further improve the paper.

Can you please do your best to address as many issues/suggestions as you can within the manuscript, and respond to the rest in a response letter. Whilst you continue to improve the quality of the manuscript, could you please keep trying to explore ways to simplify/clarify the manuscript. Both the reviewer and I are still struggling to follow the logic in some parts of the manuscript, which obscures the true quality of your study.

Looking ahead, once you have addressed the issues outlined by the expert reviewer, I will re-read the manuscript with great care, and provide feedback to improve the clarity of the manuscript - should that still be required. I expect this will take a couple more rounds of increasingly minor edits.

I hope all is well with you in this difficult time.

Genevieve

·

Basic reporting

See below (general comments).

Experimental design

See below (general comments).

Validity of the findings

See below (general comments).

Additional comments

Comments to the author:

I am happy to see that the authors decided to write a revision after all this time. I am also happy with the way the authors addressed my previous comments. However, just like test-retest reliability of psychological tests declines over time, reviewer consistency is also subject to change over time. So, having to carefully reread your interesting study, I ended up with a few additional recommendations/questions:

1.
679. I suggest to replace zwart (=CCVCC) by something else as example for CVCC – or correct CVCC into CCVCC if that was an error.

2.
1501-1503. “Familial risk was a relevant predictor for PROEF phonological awareness scores in the Belgian sample, reflected in slightly lower scores for children with a familial risk for dyslexia (β = 0.23, t = 1.34, p = .183) with a small effect size (d = 0.23).”

Two questions:
Q1. As I read it, this suggests that your model’s result is interpreted that PA was lower in familial risk than in no risk, regardless of time. But, from your model in the Table (PA at T2 as outcome, versus familiar risk and PA at T1 as regressors), and the R-markdown file (Formula: T2PF_TotalZ ~ T1CF_TotalPc + Cond + FamRisk + T1PF_TotalZ + (1 + T1PF_TotalZ | School)
) I believe this β -statistic suggests that children with familial risk improved less in PA from T1 to T2 than those without this risk (unique variance of PA T2 explained after controlling for, among others, shared variance with PA at T1).
Right?

Q2. If I understand it correctly, familial risk is included as relevant factor because it has a significant positive impact on the model fit (as indexed by the AIC). At the same time, its regression coefficient, as reported in the text, has a nonsign. p =.18. How do you reconcile these? Perhaps you can make a short comment as this seems contradicting.

3.
1505-1507. “Age was a relevant covariate in analyses of in-game response times of letter-speech sound identification (β = 0.02, t = 2.22, p = .030) and written lexical decision tasks (β = 0.20, t = 3.17, p = .002). In both cases, younger children took, on average, longer to reply than older children.”
Please add ‘in the Dutch sample’ for letter-speech sound identification, and ‘in the Belgian sample’ for written lexical decision

4.
I do advise to check the discussion to make it more in balance with the results. Now, it sometimes appears as a somewhat unbalanced capitalization on significant statistics, e.g. on letter-sound matching accuracy:
1714-1717 “Within the Belgian sample, the children who played the literacy game improved their letter knowledge more than the other two groups (as measured by the accuracy in the timed letter-speech sound identification task) and..”
1977-1979 “We found that children who played the literacy game made more pronounced progress in letter knowledge than their peers who played the math game and those who did not play any game.”

I would suggest to provide a better context for the effect on letter-sound accuracy in these discussion-sections, as it is a. only found in the Belgian sample, b. between reading game and control, not significantly so bt reading vs. math, and c. the fact the controls had a higher T1 level makes interpretation more cautious as this might also be a factor in the interaction (regression towards mean, less room for improvement)

5.
In the discussion, it remains somewhat unclear what your study contributed on Research Question 1 (in contrast to RQ2 and RQ3 where the contribution is clearly stated). Please provide a concluding remark on what your study results tell us on the research question 1. It seems to me your study provided some support for using online (in game) assessment to evaluate these training effects due it’s more fine-grained level of measurement as opposed to standardized clinical measures, but that further research into online measurement of change during gameplay is needed. Right?

6.
2297-2302. “. Due to an earlier pilot showing floor results and due to time constraints for testing at schools we decided not to collect such data at pre-test. As a result, we could not directly test interactions between reading fluency improvement and other factors, but by controlling reading fluency outcome for reading precursors at pre-test (letter knowledge, phonological awareness, rapid automatized naming and age) we are still convinced that our results are robust and meaningful.”

Although I am sympathetic to the argument that the game has promise to improve reading fluency and that some of the results has supported this, I cannot agree with stating that the results on reading fluency improvement can be related to the game in a robust way as, apart to the missing prestest,
a. randomization was clustered, that is randomized over, not within classes, and the number of classes/schools was very small, so teacher influence might be an additional factor in differences in reading fluency at posttest
b. reading fluency effects were not found in the Dutch sample, and in the Belgian sample only a significant effect was present between reading game and control, but not between math and reading game.
So, I basically suggest to delete the “robust” here.

---

## Round 0.4 · Minor Revisions

Dear Toivo,

Thank you for resubmitting your revised manuscript to PeerJ. At this stage, we have received six reviews of your paper over three review stages, which have focused primarily and the results and statistics, and I believe the methodological issues should be addressed by now.

This leaves the clarity of the manuscript. As I mentioned in my previous response, both myself and the final reviewer were still struggling to follow the logic of the manuscript in some places in the previous version, and so my goal for this review round was to check that closely, and try and provide as much support as I could for further improvements. My first reading of the manuscript indicated that some work is still required, and so to save us both time and confusion, I downloaded the latest version of the main document (ie .doc), and have provided suggestions within the manuscript itself. I can only provide a PDF version of those comments in the PeerJ system (ie not a .doc version). If you would like the .doc version (which may save your time), please contact me via email once you have received this message, so I can send it to you directly (genevieve.mcarthur@mq.edu.au)

Before you look at my suggestions, I would like to apologise for any perceived bluntness in my language in my comments in the doc. This is completely unintended. I am just trying to convey my meaning in the smallest number of words possible.

I would also like to warn you that we have a good way to go with regards to the clarity of the manuscript. You will see that I have provided a lot of suggestions about the writing, as well as the formatting of the manuscript. My suggestions are fairly standard, and so if it is not clear to you why I am suggesting what I am suggesting, I strongly recommend that you call on the support of your team - particular senior members - to help you. Some of my suggestions are fairly fundamental, and I am a little surprised your senior co-authors have not detected these issues in previous revisions.

I am very willing to support you in further improving the clarity of the manuscript, but would very grateful if you would make sure at least two of your senior co-authors do a thorough job of reading and revising your manuscript to improve its clarity and formatting before you send it back. It is really their responsibility to provide you with this level and degree of feedback and support.

I wish you the best of luck with your revisions.

Genevieve

---

## Round 0.5 · Minor Revisions

Hi Toivo,

Thank you for the revised version of your manuscript.

The clarity of your manuscript is improved, but it still needs some work to make it clear to readers. I have some “higher-level” and structural suggestions for you to improve the clarity of the manuscript in a number of places. Once these issues are addressed, we can focus on clarifying the manuscript at the word level. So, here are my initial suggestions:

1. I found parts of the manuscript hard to read because the paragraphing is inconsistent. There are gaps between some paragraphs and not others, which is making it difficult for me to follow your logic. Also, some paragraphs appears to be extremely long, which makes text appear to ramble. Can you please go through and fix the paragraphing (ie add lines between paragraphs OR remove all lines between paragraphs and use indents). And can you keep in mind that each paragraph should focus on making a single point.

2. The Statistics Methods is difficult to follow. Can you either merge this section with the results section OR provide a step-by-step guide about what statistical methods you used, by you used them, and how those statistics addressed the aims

3. The Results section is also difficult to follow. Can you please divide into sections, entitle each section with the aim of the study (you currently allude to this at some points in the Results, but you do not rephrase the aims in the same way they are phrased at the end of the Introduction), and then outline the results relevant to that aim.

4. Can you please do the same in the Discussion. That is, include separate sections for each aim, entitle each section with the aims as phrased in the Intro, and explain what the results tell us about each aim, and how that ties into previous research. Do not bring up new information (such as justifying certain methodological approaches) in the Discussion. If it is important, it should be done in the Intro or Methods.

5. I note that you are still dividing up the sample for the Belgian students and Netherlands students. This seems to “come out of nowhere” in relation to your aims. You need introduce and justify this approach either in the Introduction somewhere or the Methods.

I am hopeful that these higher-level changes will improve the manuscript to the point where the next round of suggestions focus more on more minor suggestions at the word level.

I hope this helps.

Genevieve

---

## Round 0.6 · Minor Revisions

Dear Toivo and Team,
Thanks for your patience and persistence with the review process. My name is Nic Badcock and I’ve been asked to take over as editor for your manuscript as Gen is currently in the process of moving to the other side of the country! Gen and I have worked together for many years so I hope that I can join the process without too much disruption and assist in finalising your manuscript.

Firstly, thanks for the updates in relation to Gen’s previous comments. It seems fair for me to focus on the current version of the manuscript, so this is what I’ve done. I think the section headers address previous concerns nicely. Well done on this.

Considering Gen’s comment about the wording, I’ve worked through the manuscript very carefully. I have made detailed notes about suggestions for wording changes for the first 14 pages (of the PDF) and then I have tracked further suggestions in a Word version of the document (attached as a PDF with tracked-change suggestions). I have made extensive suggestions, mostly minor, but I feel that these will improve the clarity of the expression.

If you agree with the changes, please feel free to adjust/accept without responding specifically to each suggestion in a response letter. For instances where you disagree with the suggestions or the comments are more substantial (e.g., number 30 in relation to the layout of the hypotheses), it would be helpful to provide a response explaining how you’ve handled these comments.

I do hope these provide useful. In the interest of expediating the process, I’m happy to be contacted by email outside of the review portal.

Many thanks,
Nic

Specific suggestions: Page and line numbers refer to the PDF document.
1. Page 4, line 25: “…valuable longitudinal playing data of large groups of…” > consider replacing ‘playing’ with ‘game’
2. Page 4, line 32: “…according to the DSM-5 (American Psychiatric Association, 2013).” > consider putting the dropping ‘according to the DSM-5’ from the sentence and just including (DSM-5, American Psychiatric Association, 2013) as the citation. I think the ‘according to’ can be implied from this. I found it slightly awkward at the end of the sentence.
3. Page 4: line 34/35: “…this developmental disorder affects around 4 to 12%...” > suggest replacing ‘this developmental disorder’ with ‘dyslexia’.
4. Page 4: line 44: “Dyslexia has been shown to be a disorder with a multifactorial aetiology…” > consider ‘Dyslexia has multifactorial aetiology’
5. Page 5, line 51: “The most prominent ones are…” > consider: ‘The most prominent factors are…’
6. Page 5, line 53: “Early performance on these skills has been found to predict both reading…” > consider replacing ‘has been found to predict’ with ‘predicts’
7. Page 5, line 57: “…respect to the relative weight of each of the cognitive and behavioural predictor of reading…” > these a plurality mismatch here: Please choose either (1) ‘each of the’ + ‘predictors’ (needs an ‘s’) or (2) ‘each’ + ‘predictor’ (dropping ‘of the’). I’d favour (2). Fewer words usually feels best to me.
8. Page 5, lines 58 to 60: Sentence beginning: “Letter knowledge is most predictive in…” Two points on this one. (1) It seems that a sequential/Oxford comma would be best in the final item (i.e., after ‘automatised naming’ = ‘automatised naming, and phonological awareness…). If this is the case, semicolons between items would be clearest. And (2), “…phonological awareness are important indicators in Dutch…” > suggest ‘indicators’ should be replaced with ‘predictors’ for clarity.
9. Page 5, line 64: “…which in English orthography rarely coincide…” > suggest ‘…which rarely coincide in English orthography…’ (otherwise we technically need to offset ‘in English orthography’ with commas, which feels more awkward to read.
10. Page 5, line 67: “…for long-term predictions.” > Could just be ‘prediction’ without the ‘s’ here
11. Page 6, line 69: “Instead of measuring only the availability…” > consider reversing ‘measuring only’ to ‘only measuring’
12. Page 6, line 74: “knowledge, phonological awareness and rapid automatised naming,” > consider the sequential/Oxford comma in here. Not everyone likes them, but given that they’re used for APA formatting in the citations, I think it’s best to use them throughout
13. Page 6, lines 75 to 80: “Rapid automatised naming seems more an individual characteristic than a trainable skill…” This sentence doesn’t seem to connect with the next one – it seems to stand alone. If it’s one of the ‘precursors’ targeted in the reading intervention in the next sentence, it would be helpful to make this explicit. If that’s not the case, then it feels like more needs to be added to the sentence to finish the thought.
14. Page 6, line 79: “…do not transfer to reading at the end of first grade.” > Would ‘beyond the first grade’ work here? (instead of ‘at the end of first grade’).
15. Page 6, line 83: “During the past decade, it has been shown that one promising way to deliver such an extended” > given that the reference goes back to 2008 (i.e., beyond a decade), to reduce words and increase accuracy, consider dropping the start of the sentence, beginning at ‘One promising way…’. Could also drop ‘such’
16. Page 6, line 96: “Finnish with its very transparent orthography” > could drop ‘very’ here.
17. Page 6, line 89: “The first version of the game therefore” > suggest that the ‘therefore’ needs to be offset with commas (i.e., ‘The first version of the game, therefore, aimed …’), or more preferably, used to lead the sentence (i.e., Therefore, the first version of the game
18. Page 7, lines 95-96: “…game do not only train grapheme-phoneme correspondences and phonological awareness, but also syllable and word reading fluency…” > suggest dropping ‘do not only’ and changing ‘but also’ to ‘in addition to’
19. Page 7, line 103: “…skills (McTigue, Solheim, Zimmer, Uppstad, 2019).” > missing the & in the citation here
20. Page 7, line 111: “…in a study by Saine et al. (2010) first graders…” > comma needed after (2010) here – but see next suggestion:
21. Page 7, lines 111 to 113: “Regarding reading fluency, for example, in a study by Saine et al. (2010) first graders at cognitive risk playing GraphoGame in Finnish caught up with children not at cognitive risk with respect to reading fluency in second grade.” > I wonder if this sentence could be simplified. Please consider: ‘Regarding reading fluency, for example, Saine et al. (2010) found that playing GraphoGame in Finnish improved at-risk first graders’ fluency to the level of typically-developing peers by second grade.’
22. Page 8, line 132: “…reading fluency, phonological awareness and/or letter-sound knowledge…” > suggest adding a comm before the ‘and/or’ for the sequential/Oxford comma
23. Page 8, line 137: “…speed itself.” > could drop ‘itself’ here.
24. Page 9, line 138: “…we ask the following questions:” > should be singular ‘question’ here
25. Page 9, lines 140 to 141: “… not investigated effects relating to participant characteristics yet.” > suggest: ‘not investigated participant characteristics.’
26. Page 9, line 142: “…a young age…” > suggest ‘at’ or ‘of a young age’ or simpler ‘younger participants’ – may be able to drop ‘(compared to other participants)’
27. Page 9, lines 144 to 145: “benefit more of the GraphoGame‐NL than children without any such risk factor.” > suggest ‘of’ should be ‘from’ and ‘without risk factors’ to end the sentence.
28. Page 9, line 147: “…highest game level that was reached, …” > suggest ‘highest game level achieved,’
29. Page 8, lines 150 to 151: “…for a long enough period of time…” > consider: ‘for a sufficient period of time’
30. Page 9, line 155: For questions 1 to 3, there’s a brief introduction before the question. There’s isn’t the case for Question 4. The consistent structure can be helpful for the reader. Having said that, I wonder whether it might be helpful to lead the section with:
We have four research questions:
1. …
2. …
3. …
4. …
And then provide the explanation, if needed, for each in a separate paragraph. I think this would make it easier for readers to find the questions and get a more rapid overview. Similarly, you may also want to delineate the hypotheses. In doing so, it may be the case that the explanations are unnecessary and you can simply list the research questions and then the hypotheses.
We have four research questions and hypotheses.
Research Questions:
1. …
2. …
3. …
4. …
Hypotheses:
1. …
2. …
3. …
4. …
Alternatively, they could be interleaved:
Question 1: …
Hypothesis 1: …
Question 2: … etc,
Please consider whether one of these suggestions brings more clarity to the expression.
31. Page 11, line 192: “…1990) a widely used literacy…” > comma needed after the parenthesis
32. Page 11, line 199: “…game, the reader is referred to Appendix 1.” > this could just be: ‘…game, see Appendix 1.’
33.
Page 12, line 210: “…this should not be a big problem as they are all exposed…’ > consider: ‘… this was not deemed an issue as all children are exposed…’
34. Page 12, line 217: “…training goes from…” > I’m unsure whether it’s intentional, but it’s more conventional to report in a past tense, so ‘training went from’ instead of ‘goes’. This appears in a few instances (e.g., in the next sentence “… there are inherent…’ would be ‘…there were inherent…’). Please check carefully. (Introductions can be written in future tense, though it often makes more sense for them to be past tense, reporting on what was done. Please check from line 122: “The aim of the current study is…” – ‘is’ could be ‘was’—and later, “Our secondary aims are….” > ‘Our secondary aims were…’
35. Page 13, line 229: “…playing time on task.” > consider ‘task time.’
36. Page 13, line 233: “…whereas at other locations children…” > suggest ‘…whereas, children at other locations...’ to avoid offsetting ‘at other locations’ with commas
37. Page 13, lines 234 to 235: “… games, the teachers and student assistants at least once a week asked them about their progress…” > consider: ‘…games, at least once per week, the teachers and student assistants asked about progress…’
38. Page 13, line 240: “Post-testing (T2) being conducted in November and December 2015.” > This structure doesn’t feel like a complete sentence to me. Changing the ‘being’ to ‘was’ could fix this, or leaving the wording as is, but joining it to the previous sentence with a comma after ‘November 2015.’
39. Page 13, lines 240 to 241: “The behavioural assessments at both time points took place…” > Consider: ‘Both behavioural assessments took place…’
40. Page 13, line 250: “In the pen-and-paper word reading fluency assessment at T2 students…” > comma needed after T2
41. Page 14, line 253: “…occur in the game nor in any other assessment…” < comma after ‘game’
42. Page 14, line 258: “…correlated strongly at r = 0.93, …” > suggest a colon or semicolon instead of a comm at the end here
43. Page 14, line 260: “…not statistically differ in preliminary…” < suggest reversing wording: ‘differ statistically’
Note: started making tracked changes in the Word document at this point.
44. Page 15, lines 290 to 291: “No reliability measures are provided for this test by the author.” > could these metrics be calculated and reported for the available data from this study?
45. Page 15, line 294: “…game and assessed at both T1 and T2.” > Consider dropping the references to T1 and T2 in the description of each measure, and provide a table or summary sentence/paragraph that details which measures where used when.
46. Page 16, line 319: “We therefore also intended to measure…” > regarding the word intended, consider adjusting to reflect what was done. I’m assuming this was done so consider: ‘We therefore measured …’
47. Page 17: Sample size > It would be helpful to confirm how the power analysis was conducted (e.g., using G*Power or some other recommendation)
48. Page 17, lines 336 to 337: “Our aim was to include 100 participants per group.” > Consider reporting what was achieved at this point – the aim statement could be dropped.
49. Page 18, lines 347 to 349: “The final three schools joined with one classroom and each of the three gaming conditions was again assigned to one of the classrooms.” > consider rewording this one – slightly confusing as is: ‘The final three schools had one classroom each; therefore, each school was assigned to one of the three gaming conditions.’
50. Page 19, lines 364 to 366: “To facilitate interpretation, the outcomes were centred and z-transformed where possible, so that the model coefficient β is identical to the effect size Cohen's d.” > Would you be able to include a reference for this?
51. Page 19, line 374: “A potential covariate was only kept if it reduced AIC by at least two…” > This may be a common rule-of-thumb, but is there are citation you could include for this as well?
52. Page 21, line 425: “Most notably, because of data retrieval problems…” > consider replacing ‘because of’ with ‘due to’. I wouldn’t say that it’s incorrect, but I think ‘due to’ reads more easily. There are some other instances of this you could adjust if you agree. (similarly with ‘carried out’ and ‘conducted’ – again, completely up to you. I’ve suggested it a couple of times to go with one word rather than two but I don’t want to undermine your voice in the writing, so please ignore/adopt as you see fit!)
53. Page 23, line 469: “…advanced Dutch sample but found such an effect for the CELF-IV in the Belgian children…” > here you’ve got Dutch ‘sample’ vs. Belgian ‘children’. Please consider being consistent with the sample vs. children references throughout (i.e., just pick one and use it in all instances).
54. Page 25/26, lines 508/539 : “To answer the second research question (i.e., whether there are certain subgroups of children who…” vs. “To answer the third research question whether in-game metrics would be relevant predictors for…” > please consider phrasing the questions in the same way. Either: (a) include both questions in parentheses (as for question 2) or (b) put them in text (as for question 3). (see also Page 28, line 564 – different phrasing for question 4. Can be helpful for the reader to have consistent phrasing across all common elements)
55. Page 27, line 553: “…contained two nonlinear interaction surfaces of CELF phonological awareness…” > Please check whether the use of ‘surfaces’ is correct here. I’m not familiar with this usage.
56. Discussion - Research Question: Please consider restating the research questions verbatim as part of the Discussion.
57. Page 30, line 613: “… but we did not find evidence for that either.” > Consider rephrasing: ‘but this was not evident in our findings.’
58. Page 36, lines 747 to 748: “These skills are not directly related, might require intermediate developmental steps, may take longer and be overall smaller” > a little more information would be useful here clarify the points. For example, please specify the end point of ‘might require intermediate steps’ (i.e., to achieve X), and ‘may take longer (to train?’) and be overall smaller’ (effects?).

---

## Round 0.7 · accepted · Accept

Dear Tovio and team,

Thanks to you and the team for working through all of these points. Your updates and rationale all work nicely. I appreciate that this has been a long time coming and I hope you can spare a moment to celebrate getting this over the line. Well done!

Best wishes for the future,
Nic Badcock